# *When to Think, When to Speak*:
# Learning Disclosure Policies for LLM Reasoning

**Jiaqi Wei** [1*]  **Xuehang Guo** [2*]  **Pengfei Yu** [3*]  **Xiang Zhang** [4]  **Wanli Ouyang** [5]
**Siqi Sun** [6]  **Qingyun Wang** [2]  **Chenyu You** [7]

## Abstract

In single-stream autoregressive interfaces, the same tokens both update the model state and constitute an irreversible public commitment. This coupling creates a *silence tax*: additional deliberation postpones the first *task-relevant* content, while naive early streaming risks premature commitments that bias subsequent generations. We introduce *Side-by-Side (SxS)* Interleaved Reasoning, which makes *disclosure timing* a controllable decision within standard autoregressive generation. SxS interleaves partial disclosures with continued private reasoning in the same context, but releases content only when it is *supported* by the reasoning so far. To learn such pacing without incentivizing filler, we construct entailment-aligned interleaved trajectories by matching answer prefixes to supporting reasoning prefixes, then train with SFT to acquire the dual-action semantics and RL to recover reasoning performance under the new format. Across two Qwen3 architectures/scales (MoE **Qwen3-30B-A3B**, dense **Qwen3-4B**) and both in-domain (AIME25) and out-of-domain (GPQA-Diamond) benchmarks, SxS improves accuracy–*content-latency* Pareto trade-offs under token-level proxies (e.g., inter-update waiting).

## 1. Introduction

Autoregressive large language models (LLMs) communicate through a single visible token stream (Wei et al., 2025a; Yang et al., 2026; Duan et al., 2026; Wei et al., 2025b; Duan

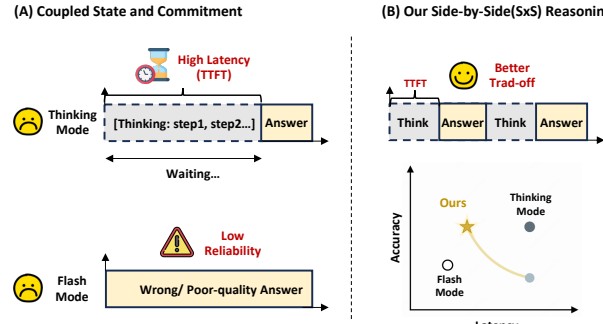

*Figure 1.* **Motivation and overview. (A)** In a single visible stream, delaying disclosure yields a long *silence tax* before task-relevant content appears, while naive early streaming can reduce delay but risks *premature commitment* that biases what follows. **(B)** SxS makes visibility controllable: the model discloses only *reasoning-supported* partial answers (*speak*) while continuing private deliberation (*think*) in the same autoregressive context, improving the accuracy–content-latency trade-off.

et al., 2025; Pan et al., 2025; Liu et al., 2025a). In this interface, each generated token simultaneously (i) updates the model's internal state and (ii) becomes a *public commitment* that fixes a visible prefix and constrains subsequent generation. This coupling is convenient but structurally limiting for interaction: users care about *when task-relevant content is disclosed with justification*, whereas the model often benefits from additional deliberation before committing to substantive claims. The resulting tension is fundamental: delaying disclosure can improve reliability but increases perceived waiting (commonly tracked by system metrics such as TTFT, though not equivalent to content latency) (Liu et al., 2025c; Gemini Team, 2025; Jiang et al., 2025), while responding immediately risks premature content that biases what follows.

Chain-of-Thought (CoT) prompting improves final accuracy by eliciting explicit intermediate reasoning (Wei et al., 2022; Zhang et al., 2025a; Wei et al., 2026; Xu et al., 2026), but it makes the tension more visible: deliberation manifests as long user-visible preambles. System-level accelerations reduce wall-clock latency (Horton et al., 2024; Liu et al., 2025b; Ruan et al., 2026), yet they leave a complementary

[1]Zhejiang University [2]College of William and Mary [3]University of Illinois Urbana-Champaign [4]University of British Columbia [5]Chinese University of Hong Kong [6]Fudan University [7]Stony Brook University. Correspondence to: Chenyu You <chenyu.you@stonybrook.edu>, Qingyun Wang <qwang16@wm.edu>.

*Proceedings of the 43rd International Conference on Machine Learning*, Seoul, South Korea. PMLR 306, 2026. Copyright 2026 by the author(s).

question unanswered: *even at a fixed compute speed, what task-relevant content *should the model **commit to** while it is still reasoning?* We focus on *justified disclosure*: early visible text should be supported by the reasoning produced so far, rather than low-information filler that merely improves measured latency.

Prior work relaxes the coupling via tagged formats and interleaving protocols (Wei et al., 2022; Xie et al., 2025), pipelined designs that separate latent reasoning from speech (Woo et al., 2025), or specialized streaming mechanisms (Tong et al., 2025). However, in the standard single-stream setting, disclosure timing is typically governed by fixed templates or heuristics, and naively incentivizing earlier output can encourage *unsupported* or *low-information* text. In short, existing approaches either (i) fix disclosure with templates/heuristics, or (ii) reward earlier output in ways that are vulnerable to filler and premature commitment. What is missing is a mechanism that makes disclosure an explicit decision variable *and* ties early visibility to a concrete support condition – the disclosed text is required to be entailed by the reasoning prefix available at that point.

We fill this gap by framing *response pacing* as a learnable control problem within single-stream autoregressive decoding. We propose *Side-by-Side (SxS) Interleaved Generation*, where the model chooses between two actions within the same token stream: *think* (non-disclosed deliberation) and *speak* (user-facing disclosure), implemented with lightweight tags. SxS does not require a second model, a separate hidden state, or specialized inference machinery. Both *think* and *speak* tokens remain in the same autoregressive context; the only change is that *visibility* becomes a controllable attribute. There is no second channel: *speak* text is a prefix of the final response that the model chooses to reveal earlier. Subsequent *think* tokens may refine and extend the response, but should not contradict earlier disclosed commitments. This enables *anytime* interaction: the model can disclose justified partial progress early, continue deliberation afterwards, and then refine or complete the response as reasoning proceeds.

A central challenge is to learn pacing without creating incentives for superficial early output. Our approach has two parts. First, we build *entailment-aligned interleaved supervision* from standard $(x, r, a)$ triples by aligning answer prefixes to reasoning prefixes, so that early disclosures are *safe to show* given the reasoning so far. Second, we train in two stages: supervised fine-tuning (SFT) teaches the dual-action semantics, and reinforcement learning (RL) recovers reasoning performance under the new format. RL is crucial because the interleaved format induces a distribution shift: SFT learns the pacing structure, while RL restores task-optimal reasoning under the new commitment constraints.

We evaluate SxS across two Qwen3 architectures (MoE and

dense), two model scales, and two complementary benchmarks: in-domain mathematical reasoning (AIME25) and out-of-domain scientific QA (GPQA-Diamond). Beyond final-task accuracy, we report token-level *content-latency* proxies that capture *when* supported user-visible progress first appears and how long users wait between updates (e.g., inter-update waiting). Across architectures, scales, and domains, SxS improves the accuracy–latency trade-off without architectural changes, showing that pacing can be learned as a controllable behavior in standard autoregressive decoding.

Conceptually, SxS turns response streaming from a formatting choice into a learned *commitment policy* with an explicit support constraint. Our contributions are as follows:

❶ *Disclosure as control under single-stream commitment.* We formalize disclosure timing as a sequential decision problem (*think* vs. *speak*) within standard autoregressive decoding, turning visibility into a controllable attribute without architectural changes.

❷ *Justified early disclosure via entailment-aligned supervision.* To avoid premature commitments and filler, we construct interleaved trajectories by aligning answer prefixes to reasoning prefixes that entail them, so that "earlier" also means "supported."

❸ *Recovering reasoning performance and learning Pareto trade-offs.* We combine SFT (to learn the dual-action semantics) with RL (to recover reasoning performance under the new format) and demonstrate improved Pareto trade-offs across architectures (MoE vs. dense), scales (30B-A3B vs. 4B), and domains (AIME25 vs. GPQA-Diamond).

## 2. Problem Formulation

### 2.1. Generation under Coupled State and Commitment

Let $x \in \mathcal{X}$ be an input and $y_{1:T} \in \mathcal{V}^T$ a generated token sequence. Standard autoregressive decoding defines

$$p_\theta(y_{1:T} \mid x) = \prod_{t=1}^{T} p_\theta(y_t \mid x, y_{1:t-1}), \qquad (1)$$

with recurrent state $h_t = f_\theta(x, y_{1:t-1})$. In the usual single-stream interface, each token is immediately user-visible. We denote the *committed transcript* after $t$ steps by

$$\Gamma_t \triangleq y_{1:t} \in \mathcal{V}^{\leq T}. \qquad (2)$$

*Coupled commitment* means state evolution and public disclosure are synchronized token-by-token:

$$h_{t+1} = f_\theta(x, \Gamma_t), \qquad \Gamma_{t+1} = \Gamma_t \oplus y_{t+1}. \qquad (3)$$

Thus, once a prefix is disclosed, later generation must remain consistent with it.

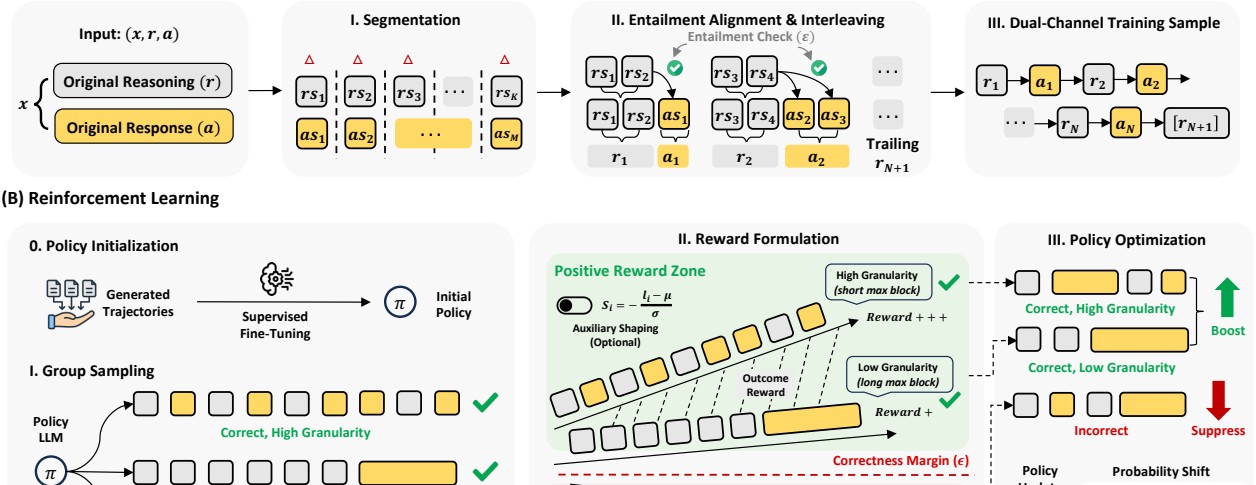

*Figure 2.* **Overview of SxS training.** We construct entailment-aligned interleaved reasoning/answer segments for dual-action SFT, then apply GRPO-based RL to *learn* the disclosure (pacing) policy.

Let Dec be a decoding rule (e.g., greedy, top-$k$, nucleus sampling) that induces a (possibly stochastic) distribution over continuations given $(x, \Gamma_t)$. We define the *decoding-feasible* set as its support:

$$\mathcal{Y}_{\text{dec}}(x, \Gamma_t) \triangleq \text{Supp}\big(\text{Dec}\big[p_\theta(\cdot \mid x, \Gamma_t)\big]\big) \subseteq \mathcal{V}^{T-t}. \quad (4)$$

Longer commitment typically makes this set more restrictive (informally, if $\Gamma_t \preceq \Gamma_{t'}$ then $\mathcal{Y}_{\text{dec}}(x, \Gamma_{t'})$ is more constrained than $\mathcal{Y}_{\text{dec}}(x, \Gamma_t)$), capturing the cost of premature public commitment.

### 2.2. Dual-Channel Autoregressive Generation

We introduce a *visibility-controlled* stream with channel actions $c_k \in \{R, A\}$ (private reasoning vs. public answer). A trajectory is $\tau = (c_1, z_1, \ldots, c_K, z_K)$ with $z_k \in \mathcal{V}$. Let $I_R(k) = \sum_{i \leq k} \mathbf{1}\{c_i = R\}$ and $I_A(k) = \sum_{i \leq k} \mathbf{1}\{c_i = A\}$, and define the projections

$$r_{1:I_R(K)} = \mathcal{P}_R(\tau), \qquad a_{1:I_A(K)} = \mathcal{P}_A(\tau), \quad (5)$$

so $T = I_A(K)$.

We separate the *private context* and *public transcript* after step $k$:

$$\Sigma_k \triangleq (x, \ r_{1:I_R(k)}, \ a_{1:I_A(k)}), \qquad \Gamma_k \triangleq a_{1:I_A(k)}. \quad (6)$$

$\Sigma_k$ conditions future generation, while $\Gamma_k$ is irreversible disclosure.

We parameterize a controlled autoregressive process as

$$p_{\theta,\phi}(\tau \mid x) = \prod_{k=1}^{K} \pi_\phi(c_k \mid \Sigma_{k-1}) \, p_\theta(z_k \mid \Sigma_{k-1}, c_k), \quad (7)$$

a conceptual factorization. In practice, $c_k$ is realized via lightweight tags predicted by the same model (no separate policy network). The updates are

$$(\Sigma_k, \Gamma_k) = \begin{cases} \big((\Sigma_{k-1} \oplus_R z_k), \ \Gamma_{k-1}\big), & c_k = R, \\ \big((\Sigma_{k-1} \oplus_A z_k), \ \Gamma_{k-1} \oplus z_k\big), & c_k = A, \end{cases}$$

$$(8)$$

where $\oplus_R$ appends only to the private stream and $\oplus_A$ appends to both. Public disclosure is monotone ($\Gamma_k \preceq \Gamma_{k+1}$). The standard interface is the special case $c_k \equiv A$.

### 2.3. Anytime Commitment as Policy Learning

We learn a channel policy $\pi_\phi(c_k \mid \Sigma_{k-1})$ that trades off deliberation and commitment. Define the *first public emission time*

$$k^\star := \min\{k \in \{1, \ldots, K\} : c_k = A\},$$

with the convention $k^\star = K + 1$ if no $A$ action occurs (in practice we ensure at least one public emission via an end-of-sequence protocol). Optimizing only $k^\star$ can reward low-information filler; instead, we measure responsiveness using a content-based statistic $g(\tau)$ that targets the onset of

*substantive* disclosed content, and we separately monitor filler rates.

**A concrete content-based latency statistic.** One instantiation used in our evaluation defines $g(\tau)$ as the onset index of the first *speak* block that contains task-relevant content (e.g., a candidate answer token), excluding a small stoplist of generic acknowledgements.

Let $\mathcal{L}_{\text{task}}(a_{1:T}, y^\star)$ be the task loss comparing the final disclosed answer stream $a_{1:T}$ to the ground-truth $y^\star$, and let $\mathcal{L}_{\text{lat}}(g(\tau))$ be a latency penalty. We optimize

$$\min_\phi \ \mathbb{E}_{\tau \sim p_{\theta,\phi}(\cdot|x)}\Big[\mathcal{L}_{\text{task}}(a_{1:T}, y^\star) + \lambda \, \mathcal{L}_{\text{lat}}(g(\tau))\Big]. \quad (9)$$

This yields an *anytime commitment policy*: the model may disclose supported partial progress early (small $g(\tau)$), continue generating private reasoning afterwards, and refine or complete $a_{1:T}$ as computation proceeds. In our instantiation, $\mathcal{L}_{\text{task}}$ is realized via an outcome-based correctness signal (used for RL), while $\mathcal{L}_{\text{lat}}$ is computed from token-level content-latency statistics.

> **PROBLEM FORMULATION**
>
> **Key idea.** Introduce a *commitment action* $c_k \in \{R, A\}$ (private reasoning vs. user-facing disclosure), making disclosure timing a learnable decision variable. We learn $\pi_\phi$ to optimize the accuracy–latency trade-off in Eq. (9), where latency is measured by a content-based statistic $g(\tau)$ (e.g., first-substantive-disclosure onset) and task quality by the final disclosed stream $a(\tau)$.

## 3. Method

We first describe how to construct supervised fine-tuning (SFT) data for dual-channel behavior from standard input–reasoning–answer triples $(x, r, a)$. We transform each triple into an *interleaved* sequence

$$S = (x, \, r^{(1)}, a^{(1)}, r^{(2)}, a^{(2)}, \ldots, r^{(N)}, a^{(N)}, [\, r^{(N+1)}\,]),$$

where each $r^{(i)}$ is a *private reasoning block* and each $a^{(i)}$ is a *user-visible answer block*, and $[\, r^{(N+1)}\,]$ is an optional trailing reasoning block. The key idea is to decide *when* a new answer block is safe to reveal. We do this by computing an *entailment-aligned boundary* sequence $\{\ell_k\}_{k=1}^{K_R}$ that maps each reasoning prefix to the largest answer prefix that is supported by the reasoning so far. We then emit answer *increments* only when $\ell_k$ increases. We then present a minor adaptation of group-based policy optimization (GRPO) for tagged, dual-channel rollouts.

### 3.1. Supervised Fine-Tuning via Entailment-Aligned Interleaving

**Segmentation.** Given $(x, r, a)$, we segment both $r$ and $a$ into blocks using a fixed delimiter. Let $\Delta \triangleq \text{``\textbackslash n\textbackslash n''}$ and define

$$R \triangleq \text{SPLIT}(r, \Delta) = [rs_1, \ldots, rs_{K_R}],$$
$$A \triangleq \text{SPLIT}(a, \Delta) = [as_1, \ldots, as_{K_A}].$$

Here $K_R \triangleq |R|$ and $K_A \triangleq |A|$ are the numbers of blocks in $r$ and $a$, respectively. In preprocessing, we normalize whitespace so that boundaries correspond to a canonical delimiter (e.g., collapsing runs of newlines to exactly two), making $\text{SPLIT}(\cdot, \Delta)$ reproducible across corpora and formatting artifacts. We also tried learned segmentation (e.g., an LLM-based $\text{SPLIT}_\theta$), but it added overhead and introduced cascading errors. We therefore use the deterministic delimiter-based segmentation throughout.

**Entailment-based alignment.** From segmentation we obtain $R = [rs_1, \ldots, rs_{K_R}]$ and $A = [as_1, \ldots, as_{K_A}]$. For each reasoning index $k \in \{1, \ldots, K_R\}$, we compute an *alignment boundary* $\ell_k \in \{0, \ldots, K_A\}$: the largest answer index such that the answer prefix $A_{1:\ell_k}$ is supported by the reasoning prefix $R_{1:k}$ under input $x$. Let $\mathcal{E}(x, R_{1:k}, A_{1:m}) \in \{0, 1\}$ be an entailment predicate, with the convention $\mathcal{E}(x, R_{1:k}, A_{1:0}) = 1$. Then

$$\tilde{\ell}_k \triangleq \max\Big\{m \in \{0, \ldots, K_A\} : \mathcal{E}(x, R_{1:k}, A_{1:m}) = 1\Big\}.$$

Because entailment checks can be noisy, we enforce monotonicity:

$$\ell_k \leftarrow \max(\ell_{k-1}, \tilde{\ell}_k), \qquad \text{with } \ell_0 \triangleq 0.$$

We also set $\ell_{K_R} \leftarrow K_A$ as a terminal safeguard to ensure the full answer is emitted even if the entailment checker is imperfect.

**Interleaving by unlocked answer increments.** We build the interleaved sequence by emitting (i) reasoning content and (ii) answer content only when new segments become safe. Specifically, whenever $\ell_k > \ell_{k-1}$, we emit the newly unlocked answer increment

$$\Delta A_k \triangleq A_{\ell_{k-1}+1:\ell_k}.$$

In practice, we also merge adjacent reasoning blocks when no new answer content is unlocked, to avoid producing overly fragmented trajectories. The full procedure is shown in Algorithm 1. Prompts for entailment detection and additional engineering details are provided in the Appendix.

**Trailing reasoning.** If the full answer becomes supported before the end of the reasoning sequence, *i.e.*, there exists

**Algorithm 1** Entailment-Aligned Interleaved Trajectory

1: **input** $x, r, a$
2: **output** Interleaved sequence $S$
3: {Step 1: Segmentation}
4: $R \leftarrow \text{SPLIT}(r, \Delta)$                 $(R = [rs_1, \ldots, rs_{K_R}])$
5: $A \leftarrow \text{SPLIT}(a, \Delta)$                 $(A = [as_1, \ldots, as_{K_A}])$
6: {Step 2: Alignment + interleaving}
7: $S \leftarrow [\,]$
8: $\ell_{\text{prev}} \leftarrow 0$
9: *open* $\leftarrow$ **true**           (start a new reasoning block)
10: **for** $k = 1$ **to** $K_R$ **do**
11:     **if** $k < K_R$ **then**
12:        $\tilde{\ell}_k \leftarrow \text{FINDMAXENTAILMENT}(x, R_{1:k}, A)$
13:        $\ell_k \leftarrow \max(\ell_{\text{prev}}, \tilde{\ell}_k)$        (monotone)
14:     **else**
15:        $\ell_k \leftarrow K_A$              (terminal safeguard)
16:     **end if**
17:     **if** *open* **then**
18:        $\text{APPEND}(S, rs_k)$
19:        *open* $\leftarrow$ **false**
20:     **else**
21:        $S_{|S|} \leftarrow S_{|S|} \oplus \Delta \oplus rs_k$
22:     **end if**
23:     **if** $\ell_k > \ell_{\text{prev}}$ **then**
24:        $\text{APPEND}(S, \text{CONCAT}(A_{\ell_{\text{prev}}+1:\ell_k}, \Delta))$
25:        $\ell_{\text{prev}} \leftarrow \ell_k$
26:        *open* $\leftarrow$ **true**
27:     **end if**
28:     **if** $\ell_k = K_A$ **and** $k < K_R$ **then**
29:        $\text{APPEND}(S, \text{CONCAT}([rs_{k+1}, \ldots, rs_{K_R}], \Delta))$
       (trailing reasoning)
30:        **break**
31:     **end if**
32: **end for**
33: **return** $S$

---

$k^\star < K_R$ such that $\ell_{k^\star} = K_A$, we append the remaining reasoning suffix as an optional trailing block:

$$r_{\text{trail}} \triangleq rs_{k^\star+1} \Delta rs_{k^\star+2} \Delta \cdots \Delta rs_{K_R}.$$

This suffix often contains self-checks after the answer is derived; preserving it encourages the dual-channel model to retain such post-solution behaviors.

**Mismatched reasoning-answer order.** Some samples present the answer in an order that does not match the original reasoning. For example, an early answer block may reveal the final result while details appear only later. In such cases, $\ell_k$ can jump early, and the interleaving can collapse toward a near-standard reasoning–response structure. We do not impose extra constraints (e.g., penalties on rapid growth of $\ell_k$ or reordering of $A$) in this work, and leave these extensions to future work.

## 3.2. Reinforcement Learning

After SFT teaches the dual-action (*think/speak*) format, we apply reinforcement learning (RL) for two goals: (i) restore task accuracy under the interleaved distribution shift, and (ii) improve the accuracy–latency trade-off. We use Group Relative Policy Optimization (GRPO) with an *outcome-only* reward as the default. To stabilize learning, we apply a simple *group filter* that removes low-signal groups (e.g., all-correct or all-incorrect). We additionally study an *optional* correctness-preserving shaping term (Appendix §C) as an ablation in §4.4.

**Tagged rollouts and parsing.** Let $\mathcal{D}$ be a prompt distribution over $x \in \mathcal{X}$. The post-SFT model defines a reference policy $\pi_{\text{ref}}$, and we optimize $\pi_\theta$ initialized as $\pi_\theta \leftarrow \pi_{\text{ref}}$. For each $x \sim \mathcal{D}$, we sample a group of $G$ tagged rollouts:

$$y_{1:T}^{(i)} \sim \pi_\theta(\cdot \mid x), \qquad y_t^{(i)} = (v_t^{(i)}, c_t^{(i)}) \in \mathcal{V} \times \{R, A\}, \tag{10}$$

where $R$ and $A$ denote *think* and *speak*. We obtain the user-visible answer by removing $R$-tagged tokens:

$$a_i \triangleq \mathcal{P}\left(y_{1:T}^{(i)}\right), \tag{11}$$

and compute a binary outcome label $g_i \triangleq g(a_i) \in \{0, 1\}$ via exact answer checking.

**Default reward (outcome-only).** Our main setting uses only the final-task outcome:

$$R_i \triangleq g_i \in \{0, 1\}. \tag{12}$$

This reward contains no explicit structural incentives, but in practice the interleaved format is largely preserved, and accuracy typically improves faster under this simple objective (see §4.4).

**Optional shaping for interleaving granularity.** We also study an auxiliary shaping mechanism to test whether interleaving granularity can be *actively controlled* without weakening the correctness signal. The shaping prefers shorter $R$-blocks (higher granularity), while enforcing a strict separation between correct and incorrect samples. Empirically, it increases granularity but slows down accuracy recovery (§4.4).

Let $b_{i,1}, \ldots, b_{i,K_i}$ denote the contiguous $R$-tagged reasoning blocks in rollout $i$, and define the maximum block length

$$\ell_i \triangleq \max_{k \in \{1, \ldots, K_i\}} |b_{i,k}|. \tag{13}$$

We define a structural score that is only informative for correct samples:

$$S_i \triangleq -\frac{\ell_i - \mu_\ell}{\sigma_\ell} \mathbf{1}\{g_i = 1\} - S_{\min} \mathbf{1}\{g_i = 0\}, \tag{14}$$

where $(\mu_\ell, \sigma_\ell)$ are computed over the correct samples within the group (or batch), and $S_{\min} > 0$ assigns a worst-case score to incorrect rollouts.

To convert $\mathbf{S} \in \mathbb{R}^G$ into final rewards $\mathbf{R} \in \mathbb{R}^G$ while preserving correctness separation, we solve the following convex quadratic program:

$$\min_{\mathbf{R} \in \mathbb{R}^G} \quad \sum_{i=1}^{G} (R_i - S_i)^2 \tag{15}$$

$$\text{s.t.} \quad R_i \geq \bar{R} + \epsilon, \ \ \forall i : \ g_i = 1, \tag{16}$$

$$R_i \leq \bar{R} - \epsilon, \ \ \forall i : \ g_i = 0, \tag{17}$$

where $\bar{R} \triangleq \frac{1}{G} \sum_{i=1}^{G} R_i$ and $\epsilon > 0$ is a fixed margin. These constraints guarantee that every correct rollout receives positive group-relative signal and every incorrect rollout receives negative signal, while still ranking correct rollouts by granularity.

**GRPO update and group filtering.** Given rewards $\{R_i\}_{i=1}^{G}$, we compute group statistics

$$\mu_x \triangleq \frac{1}{G} \sum_{i=1}^{G} R_i, \tag{18}$$

$$\sigma_x \triangleq \sqrt{\frac{1}{G} \sum_{i=1}^{G} (R_i - \mu_x)^2 + \epsilon_{\text{num}}}, \tag{19}$$

and advantages $A_i \triangleq (R_i - \mu_x)/\sigma_x$. We then apply a standard GRPO policy-gradient update with KL regularization to $\pi_{\text{ref}}$.

Groups with near-degenerate rewards (e.g., all correct or all incorrect) have $\sigma_x \approx 0$ and produce low-signal updates. We therefore drop such groups before the backward pass (used in all RL runs). When shaping is enabled, degenerate groups can also make Eqs. (16)–(17) infeasible or uninformative; we drop them in that case as well.

# 4. Experiments

## 4.1. Training Details

**Model Architectures and Initialization.** We study two models from the Qwen3 family: the Mixture-of-Experts (MoE) **Qwen3-30B-A3B** and the dense **Qwen3-4B**. Unless otherwise stated, we initialize from their *post-trained* checkpoints rather than base models. This choice preserves existing instruction-following and reasoning behaviors and reduces the amount of additional data needed to learn pacing. In a pilot comparison, applying our SFT pipeline on **Qwen3-30B-A3B-Base** results in below $40\%$ accuracy on AIME25 under Standard CoT prompting, while the post-trained **Qwen3-30B-A3B** starts at $76.9\%$.

**Supervised Fine-Tuning (SFT).** We build an SFT corpus by aggregating and deduplicating samples from Deep-Math (He et al., 2026), OpenMathReasoning (Moshkov et al., 2025), and OpenThoughts (Guha et al., 2025), yielding about 330k unique $(x, r, a)$ triples (prompt, reasoning, response). Reasoning traces and responses are synthesized with GPT-OSS-120B and filtered by outcome correctness to improve quality. We then apply Algorithm 1 to convert each triple into our interleaved dual-channel format. Training uses a global batch size of 2,048 (before sequence packing) with a maximum packed length of 32,768 tokens.

**Reinforcement Learning (RL).** After SFT, we further optimize pacing with Group Relative Policy Optimization (GRPO). We use the DAPO dataset (Yu et al., 2026) with 17k prompts, a group size of $G = 16$, and a prompt batch size of 32. To improve stability, we apply a simple variance-based filter: we skip groups where all sampled outputs are either correct or incorrect, since such groups provide little relative training signal.

**Implementation.** All experiments are run with the Slime framework (Zhu et al., 2025). Slime uses SGLang for high-throughput rollout generation and Megatron for distributed training. During SFT, we bypass rollout generation and directly train on the preprocessed interleaved samples; during RL, we use SGLang to generate rollouts for GRPO updates.

## 4.2. Evaluation

> **LATENCY METRICS**
>
> **Average Response Index (ARI).** Let $t$ be the (1-indexed) position of a token in the full sequence. ARI is the mean position index of all *speak* tokens. Lower ARI indicates that visible content is concentrated earlier.
>
> **Average Block Onset (ABO).** A *speak* block is a maximal consecutive span of *speak* tokens. For each block, we record the position index of its first token, and ABO is the mean of these onset indices. Lower ABO indicates that new visible chunks begin earlier.
>
> **Average Inter-Response Wait (AIRW).** Between two consecutive *speak* blocks, the model generates a *think* span. AIRW is the mean length (in tokens) of these intervening *think* spans. Lower AIRW indicates shorter gaps between visible updates.

**Benchmarks.** We evaluate in-domain mathematical reasoning on **AIME25**. For each problem, we sample $k = 16$ independent generations and report the average correctness across samples. To test out-of-domain generalization, we evaluate on **GPQA-Diamond** (Rein et al., 2024), which covers biology, chemistry, and physics. For GPQA-Diamond,

we sample $k = 3$ generations per question and report average accuracy. We further include **LiveCodeBench (LCB)** (Jain et al., 2025) to assess code reasoning performance, where we report `pass@1`, and **KOR-Bench** (Ma et al., 2024) to evaluate knowledge-orthogonal reasoning under rule-based and low-knowledge settings, where we report overall accuracy.

**Content-Latency Metrics.** We measure user-perceived responsiveness using token-level *content-latency* metrics computed on the full generated sequence (including both *think* and *speak* tokens). These metrics are *proxies* for perceived waiting: they capture *when* substantive visible content appears in the sequence, independent of system throughput.

### 4.3. Main Results

Our results suggest that the perceived trade-off between deliberation and responsiveness is strongly shaped by the *single-stream disclosure convention* used at deployment time. Under the standard "think-then-speak" interface, intermediate progress is often withheld until late in the trace, so longer reasoning manifests as longer visible silence. SxS changes only what is *shown*, not what is *available to condition future tokens*: the model can continue generating internal deliberation while selectively disclosing user-facing content when it is ready. In this section, we summarize what improves, where the gains come from, and what remains unchanged.

**Breaking the Silence Tax.** The most consistent effect of SxS is a redistribution of visible updates over the generation trajectory. As shown in Table 1, interleaving introduces a small overhead in total tokens (due to tagging and switching), but it substantially shortens the gaps between successive visible chunks. For Qwen3-4B, the Average Inter-Response Wait (AIRW) decreases from 21,316 tokens (Standard CoT) to 8,519 tokens (SxS, RL Final). Interpreted as a token-level proxy for user waiting, this corresponds to fewer long "silent" stretches and more frequent partial disclosures. Importantly, this improvement is concentrated in *inter-update wait* (AIRW) rather than uniformly shifting all response tokens earlier (e.g., ARI/ABO may change less), indicating that SxS primarily changes the *pacing* of disclosure.

**The Alignment Tax and RL Recovery.** Figure 3 shows a consistent "dip-and-recover" pattern. SFT teaches the model the dual-action format (alternating *think* and *speak*), but it also changes the sequence distribution: reasoning is no longer a single contiguous span. This distribution shift can reduce final-task accuracy immediately after SFT (e.g., 30B-A3B drops to $50.8\%$ post-SFT). We view this drop as a training alignment issue: the model has learned *how* to

interleave, but not yet *how to remain correct* under the new format. Outcome-based RL then acts as a corrective stage, recovering accuracy while largely preserving the interleaved behavior. Empirically, the RL stage does not simply collapse back to standard "one-shot" CoT; the interleaved structure remains present while accuracy improves. This supports the practical claim that interleaving and correctness are compatible in a single-stream model, but typically require an explicit recovery stage.

**OOD Robustness and Catastrophic Forgetting.** On GPQA-Diamond, Standard CoT after math-focused RL shows a large performance drop relative to the Base model (e.g., Qwen3-4B from $55.9\%$ to $19.0\%$), consistent with catastrophic forgetting under domain-skewed post-training. SxS retains substantially higher OOD accuracy (e.g., $49.3\%$). We treat this as an empirical observation; one plausible explanation is that dual-channel supervision couples intermediate reasoning to visible commitments, which may reduce the freedom to produce unsupported "reward-hacking" rationales that do not track the final answer. However, this mechanism is not directly measured in our current evaluation, and we leave a more targeted analysis of failure modes (e.g., unsupported early disclosures, rationale–answer inconsistency) to future work.

### 4.4. Does RL Degrade Reasoning Granularity

Here we run an additional ablation with an auxiliary correctness-preserving shaping incentive (Appendix §C) to study controllability of interleaving granularity. In our reinforcement learning experiments with GRPO, the standard reward function is outcome-based and contains no explicit structural incentives to maintain the dual-channel interleaved format. Consequently, a potential concern is that outcome-only RL provides no explicit incentive to maintain fine-grained interleaving. In the extreme, the policy could maximize reward by reverting to a single contiguous reasoning block (standard CoT), thereby discarding the pacing behavior learned during SFT. We therefore test whether interleaving granularity is stable under GRPO, and whether it can be controlled when desired.

We compare two RL runs on Qwen3-4B: (1) **unconstrained RL**, using only outcome-based rewards, and (2) **RL with an auxiliary granularity incentive**. The incentive is implemented via a correctness-preserving reward reshaping step (Appendix §C), formulated as a quadratic program that encourages shorter maximum *think* spans while constraining the sign of advantages so that correct rollouts remain favored over incorrect ones.

Figure 4 tracks the number of reasoning blocks over the first 400 RL steps. Without incentives, the average number of reasoning blocks decreases from 6.0 to 3.7, indicating some pressure toward coarser interleaving. Crucially, we do not

*Table 1.* **Main Results.** Comparison of Accuracy and Latency on AIME25 (Math) and GPQA Diamond (Science). **Color Coding**: Blue columns indicate Accuracy (higher is better), Red columns indicate Latency (lower is better). **ARI**: Avg Response Index, **ABO**: Avg Block Onset, **AIRW**: Avg Inter-Response Wait. Latency values are in token indices. **Key Finding**: On the 4B model, the Interleaved paradigm significantly outperforms standard CoT on both benchmarks, preventing catastrophic forgetting on GPQA.

| | AIME25 (Math) | | | | GPQA Diamond (Science) | | | |
|---|---|---|---|---|---|---|---|---|
| | Acc ↑ | Latency ↓ | | | Acc ↑ | Latency ↓ | | |
| **Method** | **(%)** | **ARI** | **ABO** | **AIRW** | **(%)** | **ARI** | **ABO** | **AIRW** |
| *Qwen3-30B-A3B* | | | | | | | | |
| Base Model | 74.0 | 16,942 | 16,942 | 16,942 | **64.0** | 7,203 | 6,768 | 6,768 |
| Standard CoT (SFT) | 76.9 | 17,287 | 16,958 | 16,958 | 26.1 | 14,817 | 14,502 | 14,502 |
| Standard CoT (RL Final) | **80.6** | 16,995 | 16,709 | 16,709 | 51.4 | 13,969 | 13,773 | 13,773 |
| SxS Interleaved (SFT) | 50.8 | **7,540** | **8,819** | **3,522** | 23.1 | **7,524** | **8,578** | **3,864** |
| SxS Interleaved (RL Recovery) | 76.0 | 14,132 | 14,043 | 8,619 | 46.1 | 11,090 | 11,513 | 6,370 |
| SxS Interleaved (RL Final) | 79.2 | 16,375 | 16,322 | 13,829 | 57.1 | 12,933 | 12,834 | 9,468 |
| *Qwen3-4B* | | | | | | | | |
| Base Model | 66.3 | 17,862 | 17,339 | 17,339 | **55.9** | 8,521 | 8,108 | 8,108 |
| Standard CoT (SFT) | 62.9 | 20,807 | 20,542 | 20,542 | 17.2 | 19,379 | 19,094 | 19,094 |
| Standard CoT (RL Final) | 73.8 | 21,580 | 21,316 | 21,316 | 19.0 | 16,597 | 16,338 | 16,338 |
| SxS Interleaved (SFT) | 38.3 | **8,002** | **9,619** | **4,035** | 13.8 | **8,854** | **10,106** | **4,995** |
| SxS Interleaved (RL Recovery) | 69.4 | 14,744 | 14,981 | 6,641 | 32.5 | 13,148 | 13,372 | 7,103 |
| SxS Interleaved (RL Final) | **80.0** | 17,981 | 17,818 | 8,519 | 49.3 | 15,572 | 15,709 | 7,738 |

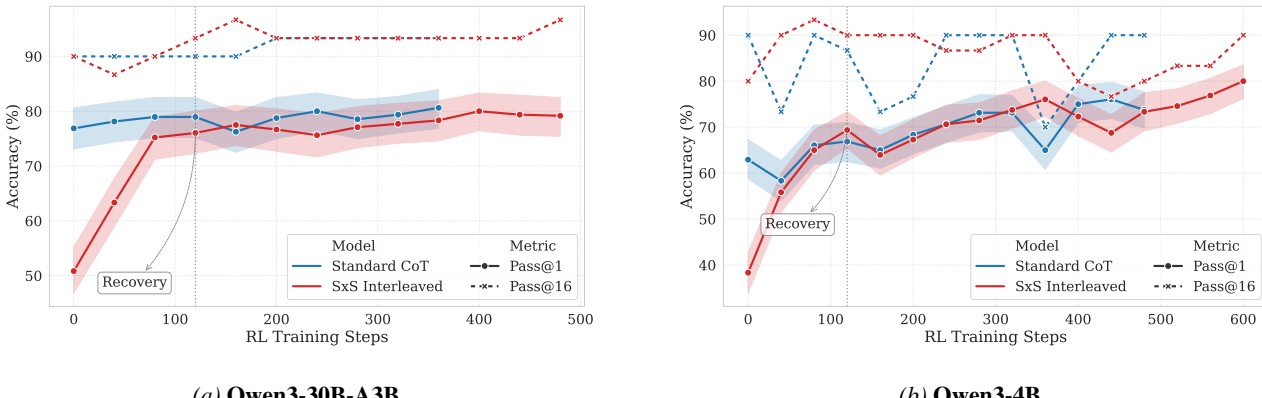

*(a)* **Qwen3-30B-A3B.**    *(b)* **Qwen3-4B.**

*Figure 3.* **RL Training Dynamics on AIME25.** We compare the Standard CoT baseline against our Interleaved Reasoning method. Shaded regions denote 95% confidence intervals. Interleaved thinking model was trained for an additional 120 steps in the RL stage to cover the recovery cost at the beginning.

observe a collapse to a single monolithic block, suggesting that the interleaved behavior is reasonably stable even under outcome-only optimization. With the incentive, granularity increases from 6.0 to 7.5, demonstrating that interleaving rate is controllable. This control comes with a cost: the incentivized run converges more slowly in accuracy, though it eventually approaches the unconstrained baseline. Overall, these results suggest that (i) outcome-only RL tends to mildly reduce interleaving frequency, (ii) the SxS format remains robust without explicit constraints, and (iii) granularity can be increased when desired via reward shaping at

some training-efficiency cost.

## 4.5. Additional analysis on LiveCodeBench and KOR-Bench

Additional results on LCB and KOR-Bench further support our core claim that SxS primarily improves the accuracy–latency trade-off, rather than optimizing raw accuracy alone. While the base models remain strongest in absolute accuracy on both benchmarks, SxS consistently yields more favorable post-training behavior than Standard CoT, espe-

*Table 2.* **Additional Results.** Comparison of Accuracy and Latency on LCB and KOR-Bench. **Color Coding**: Blue columns indicate Accuracy (higher is better), Red columns indicate Latency (lower is better). **ARI**: Avg Response Index, **ABO**: Avg Block Onset, **AIRW**: Avg Inter-Response Wait. Latency values are in token indices.

| | LiveCodeBench | | | | KOR-Bench | | | |
|---|---|---|---|---|---|---|---|---|
| | Acc ↑ | Latency ↓ | | | Acc ↑ | Latency ↓ | | |
| **Method** | **pass@1** | **ARI** | **ABO** | **AIRW** | **overall (%)** | **ARI** | **ABO** | **AIRW** |
| *Qwen3-30B-A3B* | | | | | | | | |
| Base Model | **72.23** | **8,885** | **8,715** | 8,715 | **56.56** | 1,173 | 1,163 | 1,163 |
| Standard CoT (SFT) | 54.12 | 11,079 | 10,929 | 10,929 | 22.56 | 1,580 | 1,533 | 1,533 |
| Standard CoT (RL Final) | 54.79 | 10,497 | 10,401 | 10,401 | 32.08 | 1,468 | 1,429 | 1,429 |
| SxS Interleaved (SFT) | 47.77 | 9,182 | 9,634 | **8,166** | 19.60 | **910** | **1,117** | **885** |
| SxS Interleaved (RL Recovery) | 50.90 | 10,142 | 10,221 | 9,205 | 23.44 | 1,178 | 1,293 | 1,169 |
| SxS Interleaved (RL Final) | 54.60 | 9,901 | 9,957 | 9,270 | 31.76 | 1,352 | 1,382 | 1,304 |
| *Qwen3-4B* | | | | | | | | |
| Base Model | **66.92** | **8,826** | **8,665** | 8,665 | **52.64** | 1,186 | 1,182 | 1,182 |
| Standard CoT (SFT) | 40.09 | 12,295 | 12,167 | 12,167 | 21.68 | 1,598 | 1,556 | 1,556 |
| Standard CoT (RL Final) | 39.34 | 12,685 | 12,579 | 12,579 | 19.52 | 1,715 | 1,681 | 1,681 |
| SxS Interleaved (SFT) | 26.26 | 9,036 | 10,182 | 9,027 | 22.16 | **942** | **1,151** | **960** |
| SxS Interleaved (RL Recovery) | 30.43 | 10,437 | 10,808 | 9,761 | 27.76 | 1,225 | 1,324 | 1,193 |
| SxS Interleaved (RL Final) | 39.62 | 10,841 | 10,974 | 9,631 | 32.96 | 1,438 | 1,491 | 1,321 |

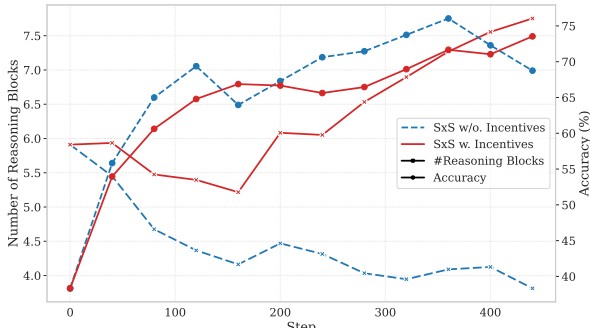

*Figure 4.* Reasoning block counts and accuracy during RL for Qwen3-4B, with and without an auxiliary incentive for interleaving granularity.

cially on Qwen3-4B. On LCB, SxS RL Final slightly improves over Standard CoT RL Final on Qwen3-4B (39.62 vs. 39.34) while substantially reducing latency, with AIRW dropping from 12,579 to 9,631; on Qwen3-30B-A3B, it reaches nearly identical accuracy (54.60 vs. 54.79) with lower AIRW (9,270 vs. 10,401). The same pattern appears on KOR-Bench: for Qwen3-4B, SxS RL Final markedly outperforms Standard CoT RL Final in overall accuracy (32.96 vs. 19.52) while also reducing AIRW from 1,681 to 1,321, and for Qwen3-30B-A3B it remains accuracy-competitive (31.76 vs. 32.08) with better latency (AIRW 1,304 vs. 1,429). Consistent with our earlier observations, the SFT-only interleaved models achieve the strongest latency reductions but incur an initial accuracy drop, whereas RL substantially recovers task performance without collapsing back to the standard one-shot CoT regime. Overall, these results show

that the benefit of SxS is robust across additional benchmarks: even when absolute accuracy gains are modest, it consistently enables earlier and denser user-visible updates under a stronger overall accuracy–latency Pareto trade-off.

## 5. Conclusion

We introduced *Side-by-Side (SxS) reasoning*, a framework that makes disclosure timing controllable in standard autoregressive decoding and mitigates the "silence tax" of single-stream CoT. By interleaving private reasoning with user-visible disclosures, SxS enables models to reveal partial progress earlier while requiring that disclosed content be supported by the reasoning prefix available at that point. Our approach combines entailment-aligned interleaved supervision with a two-stage SFT+RL pipeline. SFT teaches the dual-action semantics of reasoning and disclosure, while RL recovers performance under the new generation format. This makes it possible to improve user-visible responsiveness without reducing the problem to naive early streaming or filler generation. Across two Qwen3 architectures/scales and multiple benchmarks spanning mathematics, science, coding, and knowledge-orthogonal reasoning, SxS improves the overall accuracy–content-latency trade-off without architectural modifications. The gains are especially consistent in reducing the delay between meaningful visible updates, showing that pacing can be learned as a stable behavioral policy rather than imposed by heuristics. These findings position response pacing as a practical and underexplored control dimension for reasoning models.

## Impact Statement

This paper presents work whose goal is to advance the field of Machine Learning. There are many potential societal consequences of our work, none which we feel must be specifically highlighted here.

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

# A. Hyperparameter Configuration & Notation Reference

**Hyperparameter Configuration.** To ensure reproducibility of our experimental results, we summarize our configurations of the key hyperparameters in Table 3, spanning our dataset construction, supervised fine-tuning, and reinforcement learning stages.

*Table 3.* **Training Hyperparameters.** This table presents key hyperparameters used in our experiments, organized by training stage: model architecture specifications, SFT data construction settings, SFT training configuration, RL training parameters, evaluation protocols, and infrastructure details.

| Parameter | Value |
|---|---|
| *Model Architecture* | |
| Base Models | Qwen3-30B-A3B (MoE), Qwen3-4B (Dense) |
| Initialization | Post-trained variants |
| *SFT Data Construction* | |
| Source Datasets | DeepMath, OpenMathReasoning, OpenThoughts |
| Total Training Samples | ∼330k (deduplicated) |
| Reasoning/Response Generator | GPT-OSS-120B |
| Block Delimiter $\Delta$ | "\n\n" |
| Entailment Checker | GPT-OSS-120B |
| *SFT Training* | |
| Global Batch Size | 2,048 (before packing) |
| Max Sequence Length | 32,768 tokens |
| Sequence Packing | Enabled |
| Learning Rate | 2e-4 |
| Optimizer | AdamW |
| *RL Training (GRPO)* | |
| RL Dataset | DAPO (17k samples) |
| Group Size $G$ | 16 |
| Prompt Batch Size | 32 |
| Correctness Margin $\epsilon$ | 0.5 |
| Variance Filtering | Enabled (exclude homogeneous groups) |
| KL Regularization | Not Applied |
| Training Steps (30B-A3B) | 360 Standard CoT (+ 120 recovery steps for SxS Interleaved) |
| Training Steps (4B) | 480 Standard CoT (+ 120 recovery steps for SxS Interleaved) |
| Max Generation Length / Max Total Length | 39,000 / 40,960 |
| *Evaluation* | |
| AIME25 | $k = 16$ per problem (Average@16) |
| GPQA-Diamond | $k = 3$ per problem (Average@3) |
| Decoding Strategy | temperature=1.0, top_p=1.0 |
| *Infrastructure* | |
| Training Framework | Slime (SGLang + Megatron) |
| Inference Engine | SGLang |
| Distributed Training Backend | Megatron |

**Mathematical Notation.** To facilitate efficient comprehension of the mathematical formulations, Table 4 summarizes the core symbols and notation used in this paper.

*Table 4.* **Mathematical Notation Reference.** This table lists all mathematical symbols used in this paper, including symbols for input/output sequences, model parameters, trajectory representations, policy and generation distributions, alignment procedures, and reinforcement learning components.

| Symbol | Description |
|:---:|:---|
| $x$ | Input prompt or query |
| $y_{1:T}$ | Generated token sequence of length $T$ |
| $\mathcal{V}$ | Vocabulary set |
| $\theta$ | Token generation model parameters |
| $\phi$ | Channel policy parameters |
| $h_t$ | Recurrent hidden state at step $t$ |
| $\Gamma_t$ | Committed transcript (public output) after $t$ steps |
| $c_k$ | Channel action at step $k$ ($R$ for reasoning, $A$ for answer) |
| $z_k$ | Token emitted at step $k$ |
| $\tau$ | Complete trajectory $(c_1, z_1, \ldots, c_K, z_K)$ |
| $K$ | Total trajectory length |
| $r_{1:I_R(K)}$ | Private reasoning subsequence |
| $a_{1:I_A(K)}$ | Public answer subsequence |
| $I_R(k)$ | Count of reasoning tokens up to step $k$ |
| $I_A(k)$ | Count of answer tokens up to step $k$ |
| $\Sigma_k$ | Private context (full state) at step $k$ |
| $\pi_\phi(\cdot)$ | Channel policy distribution |
| $p_\theta(\cdot)$ | Token generation distribution |
| $k^\star$ | First commitment time (first $A$ action) |
| $g(\tau)$ | Content-based latency statistic |
| $\mathcal{L}_{\text{task}}$ | Task loss (accuracy objective) |
| $\mathcal{L}_{\text{lat}}$ | Latency penalty |
| $\lambda$ | Trade-off coefficient between accuracy and latency |
| $\Delta$ | Delimiter for block segmentation ("\n\n") |
| $R$ | Segmented reasoning blocks $[r_{s_1}, \ldots, r_{s_{K_R}}]$ |
| $A$ | Segmented answer blocks $[a_{s_1}, \ldots, a_{s_{K_A}}]$ |
| $K_R$ | Number of reasoning blocks |
| $K_A$ | Number of answer blocks |
| $\ell_k$ | Alignment boundary (max entailed answer index at step $k$) |
| $E(\cdot)$ | Entailment predicate |
| $\Delta A_k$ | Newly unlocked answer increment at step $k$ |
| $G$ | Group size for GRPO rollouts |
| $y_{1:T}^{(i)}$ | $i$-th rollout in group |
| $a_i$ | Parsed final answer from rollout $i$ |
| $g_i$ | Binary correctness label for rollout $i$ |
| $R_i$ | Reward assigned to rollout $i$ |
| $\ell_i$ | Maximum reasoning block length in rollout $i$ |
| $S_i$ | Target structure score for rollout $i$ |
| $\bar{R}$ | Mean reward across group |
| $A_i$ | Advantage for rollout $i$ |
| $\epsilon$ | Margin for correctness constraint in QP |
| $\mu_x, \sigma_x$ | Mean and standard deviation for advantage normalization |

# B. SFT Data Generation Details

### B.1. Entailment Detection Prompt

To implement the function (Algorithm 1, line 10), we employ a Large Language Model (specifically GPT-OSS-120B in our experiments) acting as a **Response Coverage Decider**.

The core challenge in segmentation is preventing "hallucinated entailment," where the model claims a reasoning step implies an answer line when it actually requires further unstated derivation. To mitigate this, our system prompt enforces a **No-New-Derivation** constraint. The prompt requires the model to identify how many pending solution blocks can be moved to the "covered" state using *only* the provided reasoning prefix.

The exact system prompt used is provided below:

---

**System Prompt: Response Coverage Decider**

```
You are a **Response Coverage Decider**.
**Problem:** {{ problem_text }}
**History:**
**Processed Thoughts:** {{ history.r }}
**Covered Responses:** {{ history.s }}
**Current**
**Current Thought:** {{ current_r }}
**Remaining Response Blocks:** {{ remaining_s }}
---
**What the inputs are**
* **Processed Thoughts**: reasoning blocks already incorporated.
* **Covered Responses**: response blocks confirmed as supported.
* **Current Thought**: next reasoning block for coverage.
* **Remaining Response Blocks**: not yet covered, as '"[BLOCK %ID] %Content"'.
* '%ID' is **0-based**. Continuation of Covered Responses.
---
**Task**
Determine how many **Remaining Response Blocks** can move into **Covered Responses**,
    using **only**: **Processed + Current Thought** and applying **no-new-derivation
    entailment**:
**Entailment rule (NO-NEW-DERIVATION)**
A block is **addable** iff its content is established by thoughts and included **without
    additional reasoning**.
**Allowed "zero-derivation" (OK):**
* Paraphrase/synonym (exact meaning); Reformatting (punctuation/grammar); Definitional
    substitution; Notation normalization (if result exists); Adding implicit background
    knowledge; Dropping dead ends.
**Not allowed (NOT addable):**
* New inference steps; Combining facts into new claims; Generalizing/dropping caveats;
    New computation; Applying theorems implicitly.
**Conservative rule:** If unsure, treat as **NOT addable**.
---
**Contiguity constraint**
Only add a **prefix**: Add 'k' blocks (0..k-1). Stop at the **first** non-addable block;
    do not skip.
---
**Output** (Return **ONLY** JSON):
```json
{ "num_blocks": k }

````
Where:
* 'k' is the number of addable blocks from the start of Remaining Response Blocks.
* 'k = 0' means add none.
* 'k = 1' means add block 0 only.
```

```
* In general, add blocks '0..k-1'.
* '0 <= k <= M' (M = number of remaining blocks).
**Return JSON only. No extra keys.
```

### B.2. Engineering Optimizations

A naive implementation of Algorithm 1 would sequentially query the LLM for each reasoning step . Given that reasoning traces can contain hundreds of steps, this sequential dependency creates a significant latency bottleneck.

We implement an asynchronous, optimistic parallelization strategy that reduces the wall-clock time from to (bounded by concurrency limits).

**1. Parallel Prefix Checks**  Instead of waiting for to be determined before calculating , we launch concurrent entailment checks for every cumulative prefix of the reasoning trace. For a reasoning trace split into segments , we simultaneously construct independent prompts. The -th prompt contains the reasoning context and the *full* solution set , asking the model to determine the coverage count .

**2. Monotonicity Enforcement**  Theoretically, entailment is monotonic: if reasoning entails answer prefix , then extended reasoning must entail at least . However, independent LLM calls may produce noisy, non-monotonic results (e.g., but ). We enforce monotonicity during post-processing. Let be the raw counts returned by the model. We compute the finalized boundaries via:

This effectively "repairs" local failures where the model fails to recognize previously established entailment in a longer context.

**3. Aggressive Cancellation**  To save computational resources, we implement an early-stopping heuristic. If a task for step returns complete coverage (i.e., ), we immediately cancel all pending tasks for indices . Due to the monotonicity principle, any reasoning step following full coverage must also imply full coverage. We synthetically assign for all cancelled tasks, significantly reducing API costs for easy samples where the solution is derived early in the trace.

## C. Incentive Design via Quadratic Programming

To encourage the model to maintain a high granularity of reasoning (i.e., frequent interleaving) without compromising the correctness of the reinforcement learning signal, we introduce an auxiliary reward shaping mechanism. Our goal is to assign a scalar reward $R_i$ to each sample $i$ in a batch of size $N$. This reward must satisfy two competing objectives:

1. **Preference Alignment:** The rewards should reflect a preference for shorter maximum reasoning block lengths (higher interleaving granularity).

2. **Correctness Constraint:** The reward structure must strictly separate correct answers from incorrect ones, ensuring that any correct rollout yields a higher reward than the average (thus a positive advantage in GRPO), and any incorrect rollout yields a lower reward than the average (thus a negative advantage in GRPO).

### C.1. Data Preprocessing

Let $y_i$ be the model response for sample $i$, and $g(y_i) \in \{0, 1\}$ be the binary correctness label. We first compute a raw structure penalty $l_i$ based on the maximum length of any single continuous reasoning block within the response:

$$l_i = \max_k \text{len}(\text{block}_{i,k}) \tag{20}$$

where $\text{block}_{i,k}$ denotes the $k$-th reasoning block in response $y_i$. For incorrect samples where $g(y_i) = 0$, we assign a worst-case penalty equivalent to the maximum observed length in the batch to avoid incentivizing formatting on incorrect answers. We then normalize these lengths to produce a target "structure score" $S_i$. Since we wish to minimize block length, we invert the normalized values:

$$S_i = -\frac{l_i - \mu_l}{\sigma_l} \tag{21}$$

where $\mu_l$ and $\sigma_l$ are the mean and standard deviation of $l$ over the valid (correct) samples in the batch.

*Table 5.* **Additional accuracy results on LCB and KOR-Bench.** We report LCB pass@1 and KOR-Bench overall/category accuracy (%).

| Model | Method | LCB | KOR-Bench | | | | | |
|---|---|---|---|---|---|---|---|---|
| | | pass@1 | overall | cipher | cfact | logic | oper | puzzle |
| **Qwen3-4B** | Base Model | 66.92 | 52.64 | 35.6 | 91.6 | 41.6 | 84.4 | 10.0 |
| | Standard CoT (SFT) | 40.09 | 21.68 | 18.0 | 30.0 | 6.8 | 49.2 | 4.4 |
| | Standard CoT (RL Final) | 39.34 | 19.52 | 13.2 | 31.6 | 3.2 | 48.0 | 1.6 |
| | SxS Interleaved (SFT) | 26.26 | 22.16 | 17.6 | 32.4 | 5.2 | 50.0 | 5.6 |
| | SxS Interleaved (RL Recovery) | 30.43 | 27.76 | 18.4 | 39.2 | 5.2 | 70.8 | 5.2 |
| | SxS Interleaved (RL Final) | 39.62 | 32.96 | 16.0 | 57.6 | 13.6 | 72.8 | 4.8 |
| **Qwen3-30B-A3B** | Base Model | 72.23 | 56.56 | 41.2 | 94.8 | 49.6 | 83.6 | 13.6 |
| | Standard CoT (SFT) | 54.12 | 22.56 | 26.4 | 24.8 | 12.8 | 42.8 | 6.0 |
| | Standard CoT (RL Final) | 54.79 | 32.08 | 33.2 | 30.0 | 19.2 | 70.4 | 7.6 |
| | SxS Interleaved (SFT) | 47.77 | 19.60 | 24.0 | 20.8 | 8.0 | 41.2 | 4.0 |
| | SxS Interleaved (RL Recovery) | 50.90 | 23.44 | 26.0 | 24.4 | 5.2 | 56.4 | 5.2 |
| | SxS Interleaved (RL Final) | 54.60 | 31.76 | 29.2 | 32.4 | 14.8 | 73.6 | 8.8 |

## C.2. Optimization Formulation

We formulate the reward assignment as a Quadratic Programming (QP) problem. We seek a final reward vector $\mathbf{R} \in \mathbb{R}^N$ that is as close as possible to the structure scores $\mathbf{S}$ in the $L_2$ sense, subject to the constraint that correct samples must have a reward significantly above the batch mean, and incorrect samples must be below it. Let $\bar{R} = \frac{1}{N} \sum_{i=1}^{N} R_i$ be the mean of the optimized rewards. We define the optimization problem as:

$$\min_{\mathbf{R}} \quad \sum_{i=1}^{N} (R_i - S_i)^2 \tag{22}$$

$$\text{subject to} \quad R_i \geq \bar{R} + \epsilon \quad \forall i, g(y_i) = 1 \tag{23}$$

$$R_i \leq \bar{R} - \epsilon \quad \forall i, g(y_i) = 0 \tag{24}$$

where $\epsilon$ is a margin hyperparameter (set to $0.5$ in our experiments). This constraint ensures that the advantages $A_i \approx R_i - V(s)$ maintain the correct sign relative to the baseline value function, strictly prioritizing correctness over structure.

## C.3. Solvability and Edge Cases

The problem is convex and can be solved efficiently using standard solvers. There is an optimal solution as long as not all samples are correct or incorrect, which is simply discarded in our GRPO training.

## D. Additional analysis on LCB and KOR-Bench.

## E. Related Work

**Single-stream reasoning and the cost of deliberation.** Multi-step reasoning in LLMs is often improved by eliciting intermediate computation, including Chain-of-Thought (CoT) (Wei et al., 2022; Kojima et al., 2022; Zhang et al., 2025a; Wei et al., 2026; You et al., 2024; 2025; Zhang et al., 2025b; Duan et al., 2026; 2025; Hu et al., 2025; Zhou et al., 2025), self-consistency (Wang et al., 2022), scratchpads (Nye et al., 2021), and structured decompositions (Zhou et al., 2022). These techniques operate under a single-stream interface where the generated prefix is both internal computation and user-visible output, offering little control over *when* substantive content appears; more deliberation typically manifests as longer visible preambles and higher perceived latency. CoT traces can also be unfaithful or post-hoc (Turpin et al., 2023; Lanham et al., 2023; Barez et al., 2025; Tutek et al., 2025; Wei et al., 2026; Cao et al., 2025). Our work targets this interface-level tension by treating *disclosure timing* as a first-class objective: the model should reveal task-relevant content earlier only when it is supported by the reasoning produced so far.

**Single-stream interleaving, refinement, and controllable disclosure.** Prior work introduces temporal structure within

*Table 6.* **Additional latency results on LCB.** Latency is measured in token indices; lower is better. ARI: Average Response Index. ABO: Average Block Onset. AIRW: Average Inter-Response Wait.

| Model | Method | Response Timing | | | Length Breakdown | | |
|---|---|---|---|---|---|---|---|
| | | ARI | ABO | AIRW | total | reasoning | response |
| **Qwen3-4B** | Base Model | 8,826 | 8,665 | 8,665 | 8,987 | 8,665 | 322 |
| | Standard CoT (SFT) | 12,295 | 12,167 | 12,167 | 12,425 | 12,167 | 257 |
| | Standard CoT (RL Final) | 12,685 | 12,579 | 12,579 | 12,791 | 12,579 | 212 |
| | SxS Interleaved (SFT) | 9,036 | 10,182 | 9,027 | 12,484 | 12,158 | 326 |
| | SxS Interleaved (RL Recovery) | 10,437 | 10,808 | 9,761 | 11,710 | 11,484 | 226 |
| | SxS Interleaved (RL Final) | 10,841 | 10,974 | 9,631 | 11,573 | 11,366 | 207 |
| **Qwen3-30B-A3B** | Base Model | 8,885 | 8,715 | 8,715 | 9,057 | 8,715 | 342 |
| | Standard CoT (SFT) | 11,079 | 10,929 | 10,929 | 11,230 | 10,929 | 301 |
| | Standard CoT (RL Final) | 10,497 | 10,401 | 10,401 | 10,594 | 10,401 | 194 |
| | SxS Interleaved (SFT) | 9,182 | 9,634 | 8,166 | 11,134 | 10,717 | 418 |
| | SxS Interleaved (RL Recovery) | 10,142 | 10,221 | 9,205 | 10,755 | 10,472 | 282 |
| | SxS Interleaved (RL Final) | 9,901 | 9,957 | 9,270 | 10,275 | 10,078 | 196 |

*Table 7.* **Additional latency results on KOR-Bench.** Latency is measured in token indices; lower is better. ARI: Average Response Index. ABO: Average Block Onset. AIRW: Average Inter-Response Wait.

| Model | Method | Response Timing | | | Length Breakdown | | |
|---|---|---|---|---|---|---|---|
| | | ARI | ABO | AIRW | total | reasoning | response |
| **Qwen3-4B** | Base Model | 1,186 | 1,182 | 1,182 | 1,192 | 1,182 | 11 |
| | Standard CoT (SFT) | 1,598 | 1,556 | 1,556 | 1,640 | 1,556 | 84 |
| | Standard CoT (RL Final) | 1,715 | 1,681 | 1,681 | 1,751 | 1,681 | 70 |
| | SxS Interleaved (SFT) | 942 | 1,151 | 960 | 1,666 | 1,519 | 147 |
| | SxS Interleaved (RL Recovery) | 1,225 | 1,324 | 1,193 | 1,598 | 1,512 | 87 |
| | SxS Interleaved (RL Final) | 1,438 | 1,491 | 1,321 | 1,644 | 1,584 | 60 |
| **Qwen3-30B-A3B** | Base Model | 1,173 | 1,163 | 1,163 | 1,184 | 1,163 | 21 |
| | Standard CoT (SFT) | 1,580 | 1,533 | 1,533 | 1,628 | 1,533 | 94 |
| | Standard CoT (RL Final) | 1,468 | 1,429 | 1,429 | 1,508 | 1,429 | 79 |
| | SxS Interleaved (SFT) | 910 | 1,117 | 885 | 1,661 | 1,488 | 173 |
| | SxS Interleaved (RL Recovery) | 1,178 | 1,293 | 1,169 | 1,626 | 1,512 | 113 |
| | SxS Interleaved (RL Final) | 1,352 | 1,382 | 1,304 | 1,545 | 1,460 | 85 |

one stream via reasoning-action alternation (ReAct) (Yao et al., 2023b), explicit branching/search (Tree-of-Thoughts) (Yao et al., 2023a), and observation separation (ReWOO) (Xu et al., 2023). Recent training (Liu et al., 2023c;a; Zhou et al., 2023; Liu et al., 2023b; 2022; Pan et al., 2025; Liu et al., 2025a; Zhao et al., 2025; Zhang et al., 2025b; Xiong et al., 2025; Liang et al., 2025; Sun et al., 2025) also explores interleaving reasoning with partial answers for responsiveness (Xie et al., 2025), alongside refinement, self-correction, and interruptibility (Liu et al., 2024; Shinn et al., 2023; Madaan et al., 2023; Akhauri et al., 2025; Wu et al., 2025). However, disclosure timing is usually template-driven or heuristic, and naive incentives for earlier output can be gamed by low-information filler or unsupported early commitments. We stay in the tagged-interleaving regime but make disclosure an explicit decision variable – a learned commitment policy $c_k \in \{think, speak\}$ – and use entailment-aligned supervision to preferentially disclose *safe-to-show* content.

**Efficiency and architectural separation of deliberation and speech.** A parallel line of work targets long-CoT inefficiency by controlling reasoning length or inducing concise rationales (Xu et al., 2025c;b; Kimi, 2025; Aggarwal & Welleck, 2025; Fatemi et al., 2025; Yuan et al., 2025; Luo et al., 2025; Wen et al., 2025a; Guo et al., 2026; Wen et al., 2025b), and by separating deliberation from communication through multi-module or pipelined systems (Xu et al., 2025a; Team, 2026; Défossez et al., 2024; Aytes et al., 2025). These approaches can reduce wall-clock latency or provide stronger isolation, but often rely on fixed module boundaries or additional system engineering. Our approach is complementary: we avoid extra modules and instead learn fine-grained pacing *within* standard autoregressive decoding, using a support-based construction

and an explicit accuracy–content-latency objective to control when intermediate progress is disclosed.

## F. Limitations

Our study has several limitations that are largely practical rather than conceptual.

**Entailment alignment cost and noise.** SxS relies on entailment-aligned supervision to ensure early disclosures are supported by the reasoning prefix. In our current instantiation, using a large entailment checker makes preprocessing expensive at scale, and the alignment can occasionally unlock segments too early or too late. However, the method only requires an approximate *prefix entailment oracle* and does not fundamentally depend on a 120B-class model. Cost and noise can be reduced by (i) replacing the checker with a smaller specialized NLI/reward model, (ii) distilling a lightweight checker from a large teacher, (iii) using a cascaded pipeline that prunes candidate boundaries with cheap heuristics before running entailment, and (iv) caching/batching incremental checks within each $(x, r, a)$ example. Our RL stage further provides robustness to moderate alignment noise.

**Objective design.** We instantiate pacing incentives with simple structural proxies (e.g., maximum *think*-block length) and outcome-based correctness rewards. Richer objectives that target user utility, *e.g.*, early disclosure of verifiable intermediate results, uncertainty-aware commitments, or task-specific notions of "substantive" content, are straightforward extensions.

