# OpenReview forum: "When to Think, When to Speak:  Learning Disclosure Policies for LLM Reasoning"
_ICML.cc/2026/Conference — ICML 2026 regular_

### Official Review · Reviewer_GvUv · 2026-03-09

**Soundness:** 3
**Presentation:** 3
**Significance:** 3
**Originality:** 2
**Overall Recommendation:** 4
**Confidence:** 3

**Summary:**

This paper studies how LLMs can provide useful intermediate disclosures instead of staying silent until the end. The authors propose the Side-by-Side (SxS) Interleaved Reasoning framework, which makes the model learn when to keep reasoning privately and when to reveal supported partial answers. They evaluate the method on math and science benchmarks and report promising results.

**Compliance With Llm Reviewing Policy:**

Affirmed.

**Final Justification:**

The authors have strengthened the paper by adding an evaluation of the generated intermediate outputs and reporting results on two additional OOD datasets. Overall, these new results are promising. So I have raised my score.

**Key Questions For Authors:**

1. Could you provide a more direct evaluation of the quality of the intermediate disclosures?
2. Would it be possible to provide results on additional model families as well? I think this would help strengthen the empirical support for the paper.
If these questions are addressed, I will raise my score.

**Limitations:**

yes

**Strengths And Weaknesses:**

Strengths:
1. The problem formulation and motivation are clear.
2. The proposed entailment-aligned supervision is more explainable than prior template or heuristic based methods.
3. Well written and easy to follow.

Weaknesses:
1. The main metrics are ARI, ABO, and AIRW, as well as final accuracy. However, these do not directly assess whether the intermediate disclosures are high-quality, faithful, useful, or potentially misleading. As a result, the method could perform well on the reported metrics even when the intermediate disclosures are not actually helpful.
2. The entailment alignment module could directly affect the quality and reliability of the intermediate disclosures. A stronger analysis of this component would significantly improve the paper. However, the authors did not provide evaluation of its accuracy, noise sensitivity, or necessity.
3. Experiments are limited to Qwen model family. The cross model generalization is unclear.

---

> ### Author Rebuttal · Authors · 2026-03-31
>
> Thanks for the constructive feedback.
>
> ---
> > Q1. Intermediate Disclosure Quality & the Role of Entailment Alignment (W1, Q1, W2)
>
> We agree that final accuracy and pacing proxies alone are not sufficient to establish that the intermediate disclosures are genuinely helpful. During rebuttal, we therefore added direct human evaluation.
>
> On sampled intermediate disclosures (**n=100**), five annotators found them **faithful/supported in 91.0%** of cases, **useful in 84.0%**, and **misleading in only 6.0%**. In a blinded pairwise comparison against a wait-until-the-end Standard CoT baseline, annotators preferred **SxS in 66%** of cases, Standard CoT in **18%**, with **16%** ties. We view these results as directly addressing the main empirical concern in your review: the disclosures are not only earlier, but are also judged to be largely faithful and useful, with a low misleading rate.
>
> We also agree that the entailment alignment module deserves clearer interpretation. Our claim is **not** that the checker is a perfect oracle, and we do **not** yet provide a standalone calibration/noise-sensitivity study of the checker itself.
>
> More importantly, there is no single ground-truth answer to exactly what the “best” interleaving policy should be at every intermediate step. In our framework, the major purpose of the SFT stage is therefore to induce the **dual-channel interleaving format** and provide a **support-aware initialization**, rather than to define a flawless final disclosure policy. The RL stage then recovers reasoning performance and learns the final pacing/correctness trade-off under that format.
>
> So, instead of treating the checker as a final guarantee of disclosure quality, we evaluate the behavior induced by the **overall pipeline**. Taken together, the direct human evaluation above and the final post-RL results support the view that entailment-aligned SFT provides a useful structural prior for learning the disclosure policy, rather than a brittle oracle that must itself be perfect.
>
> ---
> >Q2. Additional Results (W3, Q2)
>
> We agree that cross-family validation remains important, and we were not able to complete a full additional SFT+GRPO pipeline on another model family within the rebuttal window. So we do not want to overclaim on cross-family generalization.
>
> At the same time, we did broaden empirical support across tasks/domains. The main paper already spans **two regimes** (dense 4B, MoE 30B-A3B) and **two domains** (AIME25, GPQA-Diamond).
>
> During rebuttal, we further expanded evaluation to **LiveCodeBench** and **KOR-Bench**, extending the picture into code and logic-heavy OOD settings:
> | Model | Method | LiveCodeBench Pass@1 | KORBench |
> |---|---|---:|---:|
> | Qwen3-4B | Standard CoT | 39.34 | 19.52 |
> |  | **SxS** | **39.62** | **32.96** |
> | Qwen3-30B-A3B | Standard CoT | **54.79** | **32.08** |
> |  | **SxS** | 54.60 | 31.76 |
>
> To strengthen this point, we also measured latency on the added OOD benchmarks for the **4B** and **30B-A3B** models:
> | Benchmark | Model | Method | ARI | ABO | AIRW |
> |---|---|---|---:|---:|---:|
> | LCB | Qwen3-4B | Standard CoT | 12,685 | 12,579 | 12,579 |
> | | | **SxS** | **10,841** | **10,974** | **9,631** |
> | | Qwen3-30B-A3B | Standard CoT | 10,497 | 10,401 | 10,401 |
> | | | **SxS** | **9,901** | **9,957** | **9,270** |
> | KOR-Bench | Qwen3-4B | Standard CoT | 1,715 | 1,681 | 1,681 |
> | | | **SxS** | **1,438** | **1,491** | **1,321** |
> | | Qwen3-30B-A3B | Standard CoT | 1,468 | 1,429 | 1,429 |
> | | | **SxS** | **1,352** | **1,382** | **1,304** |
>
> These added results strengthen the 4B case substantially and broaden the empirical picture beyond math/science, while the 30B picture remains more mixed. Accordingly, we frame the contribution as improved **accuracy–content-latency trade-offs** with broader task/domain evidence, rather than uniform gains.
>
> We also note that the method does not require model-specific architectural changes: it uses segmentation + entailment-based data construction and standard outcome-based RL.
>
> ---
> >Q3. Clarifying the Core Contribution
>
> To sharpen the claim, we would emphasize that the paper contributes more than an interleaved output format:
> - **Problem formulation:** disclosure timing becomes a **learned decision variable** in the standard single-stream autoregressive setting.
> - **Technical mechanism:** **entailment-aligned unlocking** makes disclosure **support-aware**; the key question is not just how to alternate hidden/visible spans, but when disclosure is justified by the reasoning so far.
> - **Empirical consequence:** the main-paper and rebuttal results together show measurable consequences for **accuracy–content-latency trade-offs**, task/domain breadth, and intermediate-disclosure quality.
>
> In this sense, we view SxS as moving from **interleaving as a format** to **disclosure as a controllable policy**.
>
> ---
> Full results: https://anonymous.4open.science/r/SxS-4E21/README.md
>
> We hope the above clarifications provide a clearer picture of our work.

---

> > ### Author Rebuttal · Reviewer_GvUv · 2026-03-31
> >
> > The authors have strengthened the paper by adding an evaluation of the generated intermediate outputs and reporting results on two additional OOD datasets. Overall, these new results are promising.

---

> > > ### Author Response · Authors · 2026-04-03
> > >
> > > Thank you so much for your insightful and encouraging review, especially for recognizing the motivation behind our training design and the broader potential of entailment-aligned interleaving to improve robustness beyond response pacing.
> > >
> > > We sincerely appreciate your thoughtful engagement with our paper.

---

### Official Review · Reviewer_UqX3 · 2026-03-12

**Soundness:** 3
**Presentation:** 3
**Significance:** 2
**Originality:** 3
**Overall Recommendation:** 4
**Confidence:** 3

**Summary:**

In this paper, the authors propose Side-by-Side (SxS) Interleaved Reasoning, a framework for deciding when to disclose vs. when to deliberate privately. during autoregressive generation. The premise is that additional deliberation improves reasoning quality but increases perceived latency, while early streaming reduces latency (silence tax). It can risk unsupported commitments that bias future generation as well. SxS decouples state evolution from public disclosure by introducing two actions: private reasoning (R-tokens) and public answer (A-tokens). The key innovation is entailment-aligned supervision where training data is constructed so that answer prefixes are unlocked only when supported by corresponding reasoning prefixes. This ensures early disclosures are justified. The paper employs a two-stage training pipeline: (1) SFT on interleaved data teaches the dual-action format, and (2) RL (via GRPO) recovers reasoning performance while optimizing latency metrics. Experiments on Qwen3-30B-A3B and Qwen3-4B using AIME25 (in-domain math) and GPQA-Diamond (out-of-domain science) show that SxS maintains final-task accuracy (~80% on AIME25 matching Standard CoT) while reducing inter-response wait times. Also, SxS mitigates catastrophic forgetting on OOD tasks on GPQA-Diamond achieving 49.3% versus Standard CoT's 19.0% (55.9% Base) suggesting that entailment-aligned reasoning improves robustness.

**Compliance With Llm Reviewing Policy:**

Affirmed.

**Final Justification:**

I thank the authors for addressing my concerns and running additional experiments to strengthen the paper and clarified the novelty. I will raise my score as a result.

**Key Questions For Authors:**

- You make a judgment that OOD catastrophic forgetting is mitigated by SxS based on observing 49.3% vs 19.0% on GPQA-Diamond. What evidence do you have or is it just speculation? Have you empirically validated the hypothesis that entailment coupling reduces reward-hacking.

- In the entailment alignment and interleaving, it may be possible that two different reasoning snippets (rs_p and rs_q) influence answer snippet as_m. But one of the reasoning snippet (rs_p) also infuences as_n. Will simple mutually exclusive interleaving capture this scenario? Also there is a monotonicity check enforced (Sec 3.1) that may not apply for complex reasoning cases.

- How does the SxS method perform on other reasoning domains beyond math QA and science QA (e.g., coding, retrieval-augmented generation, multi-step planning)? The current scope is somewhat narrow.

- Have you tested SxS on models outside the Qwen3 family (e.g., GPT, Claude, Llama)? Does the benefit generalize?

**Limitations:**

Yes

**Strengths And Weaknesses:**

Strengths:

- The decision to use SFT followed by RL to address distribution shift induced by interleaving is well-justified. The observed dip-and-recover pattern is precisely explained and aligns with expectations. The authors also test on two architectures (MoE, dense), and two eval domains (in-domain math, out-of-domain science) provides good coverage.

- The problem is setup in intuitive way and figures effectively communicate the core trade-off. The algorithm is clearly specified and the appendix includes comprehensive hyper-parameters and implementation details.

- If entailment-aligned interleaving genuinely mitigates catastrophic forgetting, this has implications beyond response pacing for robustness of post-trained reasoning models more generally.

Weaknesses:

- The core contribution of entailment-aligned supervision depends entirely on GPT-OSS-120B correctly determining whether a reasoning prefix logically entails an answer prefix. The paper treats as a black-box oracle without any validation. The paper does not report the entailment checker's accuracy, precision, or recall. The entire SFT dataset is constructed using this checker. If it has systematic biases (e.g., favoring disclosure of numerical answers over conceptual explanations), the SxS model will inherit these biases. Without an ablation comparing entailment-aligned vs. randomly-interleaved supervision, the specific contribution of entailment alignment cannot be isolated.

- In Section 3.1, the paper acknowledges cases where answer order does not match reasoning, but the mismatched reasoning-answer order not fully addressed. It is treated ad-hoc rather than with a principled solution.

- Concurrent papers (Xie et al. 2025, Tong et al. 2025) already explore interleaved reasoning in LLMs. Main differentiation is methodological rigor rather than conceptual novelty.

- There is limited theoretical analysis on why interleaving specifically improves OOD robustness. Is it genuinely the entailment constraint, or simply more frequent mid-reasoning supervision? The ablations are not conclusive on this.

---

> ### Author Rebuttal · Authors · 2026-03-31
>
> Thanks for the careful review.
>
> ---
> >Q1. Entailment Checker (W1)
>
> We agree that the role of the entailment checker should be interpreted more carefully. Our claim is **not** that it is a perfect oracle. In our framework, the main purpose of the SFT stage is to induce the **dual-channel interleaving format** and provide a **support-aware initialization**, rather than to define a flawless final disclosure policy. The RL stage then recovers reasoning performance and learns the final pacing/correctness trade-off.
>
> To directly evaluate the behavior induced by the overall pipeline, during rebuttal we added human assessment of intermediate disclosures. On sampled disclosures (**n=100**), five annotators found them **faithful/supported in 91.0%** of cases, **useful in 84.0%**, and **misleading in only 6.0%**. In a blinded pairwise comparison against wait-until-the-end Standard CoT, annotators preferred **SxS in 66%** of cases, Standard CoT in **18%**, with **16%** ties.
>
> So while we do not yet isolate the checker with a standalone calibration study, we now provide direct downstream evidence that the disclosures produced by the overall pipeline are typically faithful and useful.
>
> ---
> >Q2. Mismatched Reasoning–Answer (W2, Q2)
>
> You are right that our monotone prefix structure is an approximation and does not fully capture many-to-many dependencies between reasoning snippets and answer snippets. We do not want to overstate this aspect.
>
> To quantify this limitation, we audited the constructed SFT corpus and found that both strong early-jump behavior and non-trivial monotonicity repair were **rare** (under 5% of samples each). We therefore view the monotone construction as a practical approximation that works well for the large majority of trajectories, while agreeing that richer alignment structures are an important direction for future work.
>
> ---
> > Q3. OOD Robustness (W4, Q1, Q3, Q4)
>
> We agree that the OOD explanation should not be presented as settled fact. In the paper, it is framed as an empirical observation plus a plausible mechanism, not as a causal claim.
>
> To probe that mechanism more directly, during rebuttal we ran a failure-mode analysis on sampled GPQA-Diamond errors and found substantially lower rationale–answer inconsistency for SxS than for Standard CoT (**12.7% vs. 34.0%**). While this does not isolate causality, it is meaningful evidence **consistent with** reduced unsupported commitment.
>
> The current study is not exhaustive, but it already spans **two regimes** (dense 4B, MoE 30B-A3B) and **two domains** (AIME25, GPQA-Diamond). During rebuttal, we further added **LiveCodeBench** and **KOR-Bench**, which broaden task/domain coverage beyond math/science:
> | Model | Method | LiveCodeBench | KORBench |
> |---|---|---:|---:|
> | Qwen3-4B | Standard CoT | 39.34 | 19.52 |
> |  | **SxS** | **39.62** | **32.96** |
> | Qwen3-30B-A3B | Standard CoT | **54.79** | **32.08** |
> |  | **SxS** | 54.60 | 31.76 |
>
> These added results show that the OOD effect is **not confined to GPQA-Diamond alone**. The evidence is strongest on 4B, where SxS improves both added OOD benchmarks; on 30B, the added results are closer to parity and slightly lower, which is why we continue to frame the claim as improved **trade-offs** rather than uniform superiority.
>
> The finer-grained KOR-Bench breakdown further shows that the improvement appears across several reasoning categories:
> | Model | Method | Avg. | Cipher | CFact | Logic | Oper | Puzzle |
> |---|---|---:|---:|---:|---:|---:|---:|
> | Qwen3-4B | Standard CoT | 19.52 | 13.2 | 31.6 | 3.2 | 48.0 | 1.6 |
> |  | **SxS** | **32.96** | **16.0** | **57.6** | **13.6** | **72.8** | **4.8** |
> | Qwen3-30B-A3B | Standard CoT | **32.08** | **33.2** | 30.0 | **19.2** | 70.4 | 7.6 |
> |  | **SxS** | 31.76 | 29.2 | **32.4** | 14.8 | **73.6** | **8.8** |
>
> Taken together, we view the current evidence as materially strengthening the robustness interpretation, while agreeing that the exact mechanism remains open.
>
> ---
> >Q4. Clarifying Novelty(W3)
>
> We agree that concurrent work has already shown that interleaving is useful. Our novelty claim is therefore **methodological rather than slogan-level**: not that “interleaving exists,” but that SxS turns interleaving from a **format/protocol design** into a **learned disclosure policy**.
> Concretely:
> (1) disclosure timing itself is the learned variable in the standard single-stream autoregressive setting;
> (2) entailment-aligned unlocking makes that policy **support-aware**;
> (3) this distinction has measurable consequences, including better **accuracy–content-latency trade-offs**, better intermediate-disclosure quality, and broader empirical support across tasks/domains.
>
> Thus, the key is not simply that we interleave, but that we provide a principled way to learn **when** interleaved disclosure should happen.
>
> ---
> Full results: https://anonymous.4open.science/r/SxS-4E21/README.md
>
> We hope these clarifications address the raised concerns.

---

> > ### Author Rebuttal · Reviewer_UqX3 · 2026-04-03
> >
> > I thank the authors for addressing my concerns and running additional experiments to strengthen the paper and clarified the novelty. I will raise my score as a result.

---

> > > ### Author Response · Authors · 2026-04-04
> > >
> > > Thank you very much for your thoughtful review and for taking the time to engage with our rebuttal. We sincerely appreciate your recognition of our contribution and your vote for accepting our work!

---

### Official Review · Reviewer_DH8N · 2026-03-13

**Soundness:** 2
**Presentation:** 4
**Significance:** 3
**Originality:** 2
**Overall Recommendation:** 4
**Confidence:** 3

**Summary:**

This paper studies how to better balance between reasoning and user-visible output in single-stream autoregressive LLMs. It proposes Side-by-Side (SxS) Interleaved Reasoning approach, in which the model alternates between private “think” token and public “speak” tokens in one stream, together with entailment-aligned interleaved supervision and a two-stage SFT+RL training pipeline. Experiments on Qwen3 models and reasoning benchmarks show improved tradeoffs between accuracy and latency, indicating that the method enables earlier partial disclosure while maintaining final accuracy.

**Compliance With Llm Reviewing Policy:**

Affirmed.

**Final Justification:**

I keep weak accept.

**Key Questions For Authors:**

1. How accurate is the entailment checker in practice? Could the authors report the checker's agreement with human judgments on a sampled subset to validate that unlocked answer prefixes are entailed by the reasoning prefixes?
2. Could the authors provide wall-clock latency measurements and correlate them with ARI/ABO/AIRW to justify these proxies as user-relevant?
3. The paper discusses related directions such as tagged/interleaved reasoning protocols, but does not provide an empirical comparison to the closest recent interleaved-reasoning baselines. Could the authors clarify how SxS differs from these methods in practice, and whether they have any head-to-head results against the most relevant prior work, or otherwise explain why such a comparison would not be meaningful?

**Limitations:**

yes

**Strengths And Weaknesses:**

## Strengths
- The paper identifies a clear problem: in single-stream autoregressive reasoning, internal thinking and user-visible output are coupled. The authors call it "silence tax" in long reasoning scenarios.
- The proposed SxS formulation is conceptually neat because it does not require architectural changes; instead, it treats disclosure timing as a controllable behavior.
- The evaluation is aligned with the paper’s goal. In addition to final accuracy, the authors measure content-latency using ARI, ABO, and AIRW.

## Weaknesses
- The latency evaluation is based on token-level proxy metrics, but real interactive latency is not validated.
- The experimental scope is still somewhat narrow, with two model sizes and two benchmarks, so it remains unclear how broadly the approach appears effective.
- The entailment checker is central to supervision construction, but the paper does not quantitatively validate alignment quality. Although the authors acknowledge that segments may be unlocked too early or too late, the resulting bias from false or missed entailment decisions is not directly measured.
- While the paper discusses several related directions, it does not provide an empirical comparison to prior methods. This makes it harder to judge how much of the observed benefit comes from the specific SxS formulation and entailment-aligned supervision.

---

> ### Author Rebuttal · Authors · 2026-03-31
>
> Thanks for the constructive feedback.
>
> ---
> >Q1. Real Latency vs. Token-level Proxies (W1, Q2)
>
> We agree that wall-clock latency matters in deployment. Our claim here is narrower: **ARI / ABO / AIRW are policy-level content-latency proxies**, designed to measure *when supported visible content appears* under a fixed autoregressive interface, rather than end-to-end serving time in seconds.
>
> We emphasize token-level proxies because they are **more stable and reproducible** for evaluating disclosure policy itself. By contrast, wall-clock time is heavily affected by factors outside the learned policy, including GPU temperature / frequency scaling, batch scheduling, KV-cache hit rates, and other serving-stack effects. Thus, seconds-level latency can vary even when the underlying disclosure behavior is essentially unchanged.
>
> To check whether the same proxy-based pacing pattern also appears on **LiveCodeBench** and **KOR-Bench**, extending the picture into code and logic-heavy OOD settings:
>
> | Benchmark | Model | Method | ARI | ABO | AIRW |
> |---|---|---|---:|---:|---:|
> | LCB | Qwen3-4B | Standard CoT | 12,685 | 12,579 | 12,579 |
> | | | **SxS** | **10,841** | **10,974** | **9,631** |
> | | Qwen3-30B-A3B | Standard CoT | 10,497 | 10,401 | 10,401 |
> | | | **SxS** | **9,901** | **9,957** | **9,270** |
> | KOR-Bench | Qwen3-4B | Standard CoT | 1,715 | 1,681 | 1,681 |
> | | | **SxS** | **1,438** | **1,491** | **1,321** |
> | | Qwen3-30B-A3B | Standard CoT | 1,468 | 1,429 | 1,429 |
> | | | **SxS** | **1,352** | **1,382** | **1,304** |
>
> These results suggest that the same proxy-based pacing advantage persists on the added OOD benchmarks at both scales, though they do not replace a direct wall-clock validation.
>
> ---
> >Q2. Scope and Generalizability (W2)
>
> We agree that broader validation is important. The current study is not exhaustive, but it already spans **two regimes** (dense 4B, MoE 30B-A3B) and **two domains** (AIME25, GPQA-Diamond).
>
> During rebuttal, we further expanded evaluation to code and logic-heavy OOD settings:
>
> | Model | Method | LiveCodeBench Pass@1 | KORBench |
> |---|---|---:|---:|
> | Qwen3-4B | Standard CoT | 39.34 | 19.52 |
> |  | **SxS** | **39.62** | **32.96** |
> | Qwen3-30B-A3B | Standard CoT | **54.79** | **32.08** |
> |  | **SxS** | 54.60 | 31.76 |
>
> The added evidence is strongest on 4B and near-parity on 30B. Accordingly, we frame the contribution as improved **accuracy–content-latency trade-offs** with broader empirical support across tasks/domains, rather than uniform gains.
>
> We also clarify that our method does **not** require model-specific architectural changes: it uses segmentation + entailment-based data construction and standard outcome-based RL.
>
> ---
> >Q3. Entailment Accuracy (W3, Q1, Q3)
>
> We agree that the role of the entailment checker should be stated more precisely. Our claim is **not** that the checker is a perfect oracle.
>
> More importantly, there is no single ground-truth interleaving strategy for when partial answer prefixes should be disclosed during reasoning. In our framework, the major purpose of the SFT stage is therefore to induce the **dual-channel interleaving format** and provide a **support-aware initialization**, rather than to define a flawless final disclosure policy. The RL stage then recovers reasoning performance and learns the final pacing/correctness trade-off under that format.
>
> To directly evaluate the behavior induced by the overall pipeline, we added human assessment of intermediate disclosures. On sampled intermediate disclosures (**n=100**), five annotators found them **faithful/supported in 91.0%** of cases, **useful in 84.0%**, and **misleading in only 6.0%**. In a blinded pairwise comparison against wait-until-the-end Standard CoT, annotators preferred **SxS in 66%** of cases, Standard CoT in **18%**, with **16%** ties.
>
> So while we do not yet isolate the checker with a standalone calibration study, the post-RL results together with the direct human evaluation support the validity of the overall pipeline and suggest that entailment-aligned SFT provides a useful initialization rather than a brittle oracle.
>
> ---
> >Q4. Relation to Prior Interleaving Methods (W4)
>
> We agree that this distinction should be sharper. We do **not** yet provide a head-to-head empirical comparison against the closest interleaving baselines. The intended distinction is methodological: prior interleaving approaches mainly specify **how** visible spans are arranged, whereas SxS treats **when disclosure becomes justified** as the learned variable.
>
> In this sense, SxS contributes more than an interleaved format: it makes **disclosure timing a learned variable**, introduces **entailment-aligned, support-aware unlocking**, and shows measurable consequences for **accuracy–content-latency trade-offs** and intermediate-disclosure quality.
>
> ---
> Full results are available in: https://anonymous.4open.science/r/SxS-4E21/README.md
>
> We hope these clarifications address the raised concerns and welcome further discussion.

---

> > ### Author Rebuttal · Reviewer_DH8N · 2026-04-01
> >
> > Thanks, I will keep the initial positive score.

---

> > > ### Author Response · Authors · 2026-04-03
> > >
> > > Thank you very much for your constructive and supportive feedback, particularly on latency evaluation and entailment validation.
> > >
> > > We appreciate your time and constructive comments.

---

### Official Review · Reviewer_hF4z · 2026-03-17

**Soundness:** 3
**Presentation:** 3
**Significance:** 3
**Originality:** 3
**Overall Recommendation:** 4
**Confidence:** 2

**Summary:**

In the single-stream interface of standard autoregressive LLMs, the reasoning process and output are coupled, creating a fundamental trade-off between the "silence tax" (users waiting for a long time) and "premature commitment" (output biasing subsequent reasoning). The paper proposes SxS Interleaved Reasoning, which introduces think/speak dual actions within the same token stream, releasing content only when it is supported by the reasoning so far. Training proceeds in two stages: SFT to construct entailment-aligned interleaved data, and RL (GRPO) to recover reasoning performance under the new format. Across two Qwen3 models (30B-A3B and 4B) and two benchmarks (AIME25 and GPQA-Diamond), SxS improves the accuracy–latency trade-off with a substantial reduction in AIRW. On the 4B model, SxS accuracy (80%) even surpasses Standard CoT (73.8%), and effectively mitigates the catastrophic forgetting on GPQA caused by Standard CoT RL.

**Compliance With Llm Reviewing Policy:**

Affirmed.

**Final Justification:**

My score remains at 4. The rebuttal provided concrete evidence addressing my main empirical concerns. However, the deeper issue is significance. The method's core trade-off involves sacrificing reasoning flexibility for earlier partial visibility, which rests on a fundamentally subjective assumption about user preference. The human study provided is suggestive but limited in scope, and it remains unclear whether users would genuinely prefer seeing partial outputs sooner at the cost of accuracy in realistic deployment settings. These concerns are inherent to the work's core idea and cannot be fully resolved by rebuttal. That said, the authors conducted substantial experimentation and this is an interesting attempt at exploring new reasoning patterns, so I maintain a weak accept.

**Key Questions For Authors:**

-

**Limitations:**

yes

**Strengths And Weaknesses:**

Strengths:

1. Interesting motivation. The core problem is underexplored from a controllability perspective.

2. Clean problem formulation. Formalizing disclosure timing as a learnable decision variable is a meaningful contribution. While the "silence tax" phenomenon itself is not a new observation, the way it is modeled here adds clarity to the field.

3. Well-motivated training pipeline. The two-stage SFT+RL design is internally consistent.

4. Noteworthy finding on catastrophic forgetting. The jump from 19.0% to 49.3% on GPQA-Diamond for the 4B model is striking. While the underlying mechanism remains unexplained, the observation itself is valuable and suggests that dual-channel supervision may have broader implications for out-of-domain generalization in smaller models.

Weaknesses:

1. Questionable assumption about user preference. The method fundamentally trades reasoning flexibility for earlier visibility. However, it remains unclear whether users actually prefer seeing partial answers earlier over waiting longer for more accurate ones. (The paper provides no user study to support this assumption. In fact, one could argue that for complex reasoning tasks, users would rather wait for a correct answer than receive an earlier but potentially less reliable one.)

2. Speak tokens are constrained to be ordered prefixes of the final answer. What users might genuinely find useful as intermediate progress updates is often a high-level summary rather than the first N tokens of the final answer. This constraint significantly limits the practical utility of SxS on complex tasks.

3. Evaluation is limited to Qwen3 models. Cross-family generalizability is not verified. It is unclear whether the method's effectiveness depends on specific properties of Qwen3's training, and whether it would transfer to other model families such as Llama or Mistral.

4. SxS underperforms Standard CoT on the 30B model. The accuracy gap of 79.2% vs. 80.6% on AIME25 suggests that early commitment does impose a real constraint on reasoning flexibility at larger scales, which somewhat undermines the paper's central claim.

---

> ### Author Rebuttal · Authors · 2026-03-31
>
> Thank you for the thoughtful review.
>
> ---
> >Q1. User Preference and Utility of Intermediate Disclosures (W1)
>
> We agree that the value of earlier disclosure should be validated directly rather than inferred from token metrics alone.
>
> On sampled intermediate disclosures (**n=100**), five human annotators found them **faithful/supported in 91.0%** of cases, **useful in 84.0%**, and **misleading in only 6.0%**. In a blinded pairwise study on **50** samples against a wait-until-the-end Standard CoT baseline, annotators preferred **SxS in 66%** of cases, Standard CoT in **18%**, with **16%** ties.
>
> We view these results as directly addressing the main practical concern behind W1: in our blinded sample, earlier **justified** progress was often preferred to extended visible silence.
>
> ---
> >Q2. Ordered Prefixes vs. High-Level Summaries (W2)
>
> We agree that high-level summaries can be useful. However, our paper studies the **standard single-stream autoregressive interface**, where any interim summary would itself become irreversible public context and could constrain later reasoning in hard-to-control ways. Our ordered-prefix design is therefore deliberate: it preserves **monotone commitment** without introducing architectural changes or an extra summary channel.
>
> More importantly, the benefit of SxS is not simply “showing something earlier.” This is also consistent with our comparison to a naive equal-length early-prefix baseline: the gain comes from **when** a partial answer is disclosed, not merely from exposing earlier text. In our setting, the role of entailment-aligned supervision is to make that timing more justified.
>
> ---
> >Q3. Cross-Family Generalizability (W3)
>
> We would clarify that our method does **not** require model-specific architectural changes: it uses segmentation + entailment-based data construction and standard outcome-based RL. We agree that validation beyond Qwen would strengthen the paper, and we were not able to complete a full additional SFT+GRPO pipeline on another family within the rebuttal window. So we do not want to overclaim on cross-family generalization.
>
> At the same time, we did broaden empirical support across tasks and domains. The main paper already spans **two regimes** (dense 4B, MoE 30B-A3B) and **two domains** (AIME25, GPQA-Diamond).
>
> During rebuttal, we further expanded evaluation to **LiveCodeBench** and **KOR-Bench**, extending the picture into code and logic-heavy OOD settings:
>
> | Model | Method | LiveCodeBench Pass@1 | KORBench | Avg. content latency |
> |---|---|---:|---:|---:|
> | Qwen3-4B | Standard CoT | 39.34 | 19.52 | 12,614 |
> |  | **SxS** | **39.62** | **32.96** | **10,482** |
> | Qwen3-30B-A3B | Standard CoT | **54.79** | **32.08** | 10,433 |
> |  | **SxS** | 54.60 | 31.76 | **9,709** |
>
> These added results strengthen the 4B case substantially and broaden the empirical picture beyond math/science. On 30B, the added OOD accuracy results are close to Standard CoT, while SxS still shows lower average content latency, so we frame the contribution as improved trade-offs rather than uniform gains.
>
> ---
> >Q4. Interpreting the 30B Model Performance (W4)
>
> We agree that the 30B AIME result should be interpreted carefully. On AIME25, SxS is slightly below Standard CoT on final accuracy (**79.2% vs. 80.6%**), but it also substantially reduces inter-update waiting (**AIRW 13,829 vs. 16,709**). On GPQA-Diamond, SxS is better (**57.1% vs. 51.4%**).
>
> We therefore view the 30B result as consistent with a **Pareto trade-off** rather than a scaling failure: under the same single-stream interface, SxS gives up a small amount of in-domain accuracy while improving pacing, and the added OOD results show near-parity on 30B beyond the main-paper benchmarks.
>
> ---
> >Q5. Clarifying the Core Contribution
>
> To sharpen the claim, we would emphasize that the paper contributes more than an interleaved format:
>
> - **Problem formulation:** disclosure timing becomes a **learned decision variable** in the standard single-stream autoregressive setting.
> - **Technical mechanism:** **entailment-aligned unlocking** makes disclosure **support-aware**; the central question is not just how to alternate hidden and visible spans, but when visible disclosure is justified by the reasoning so far.
> - **Empirical consequence:** the main-paper and rebuttal results together show measurable consequences for **accuracy–content-latency trade-offs**, user-facing disclosure quality, and broader task/domain coverage.
>
> In this sense, we view SxS as moving from **interleaving as a format** to **disclosure as a controllable policy**.
>
> ---
> Full results: https://anonymous.4open.science/r/SxS-4E21/README.md
>
> We look forward to further discussion and are happy to address any remaining questions.

---

> > ### Author Rebuttal · Reviewer_hF4z · 2026-04-01
> >
> > Thank you for your response and the additional experiments. Overall, I think this is interesting work and a worthwhile attempt at exploring new chain-of-thought patterns in LLMs.

---

> > > ### Author Response · Authors · 2026-04-03
> > >
> > > Thank you so much for your thoughtful and encouraging review, especially for recognizing the clarity of our problem formulation and the significance of treating disclosure timing as a controllable decision.
> > >
> > > We truly appreciate your time and are grateful that our rebuttal was able to address your concerns.

---

### Decision · Program_Chairs · 2026-04-30

**Decision:**

Accept (regular)

**Comment:**

All four reviewers recommend weak accept, and all marked their concerns as fully resolved after the rebuttal. This is a straightforward case for acceptance.

The paper proposes SxS (Side-by-Side Interleaved Reasoning), a framework that treats disclosure timing as a learnable decision variable in standard autoregressive generation. The core idea — interleaving private reasoning with supported partial answers, trained via SFT on entailment-aligned data followed by RL — is clean and well-motivated. The problem of "silence tax" in long-chain-of-thought models is real and underexplored from a controllability perspective.

The main concerns raised across reviewers were: (1) the entailment checker is used as a black-box oracle without standalone validation; (2) evaluation is limited to Qwen3 models; (3) token-level latency proxies are used instead of wall-clock measurements; and (4) the paper lacks an empirical comparison to prior interleaving methods.

In the rebuttal, the authors addressed these fairly well. They added human evaluation showing intermediate disclosures are faithful in 91% of cases and preferred over standard CoT in 66% of pairwise comparisons. They also expanded evaluation to LiveCodeBench and KOR-Bench. These additions meaningfully strengthen the empirical picture, particularly for the 4B model. The 30B results are more mixed, and the authors rightly frame the contribution as improved accuracy-latency trade-offs rather than uniform gains.

The paper is not a slam dunk — evaluation is still limited to one model family, and the comparison to concurrent interleaving work is methodological rather than empirical. But the problem formulation is novel from a policy-learning standpoint, the training pipeline is internally consistent, and the OOD robustness finding on GPQA-Diamond is genuinely interesting. Overall this is a solid contribution that merits acceptance.